# Diffusive hydrodynamics of hard rods from microscopics

**Friedrich Hübner**[1*]**, Leonardo Biagetti**[2]**, Jacopo De Nardis**[2]**, Benjamin Doyon**[1]

**1** Department of Mathematics, King's College London, Strand WC2R 2LS, London, U.K.
**2** Laboratoire de Physique Théorique et Modélisation, CNRS UMR 8089, CY Cergy Paris Universite, 95302 Cergy-Pontoise Cedex, France

* friedrich.huebner@kcl.ac.uk

## Abstract

We derive exact equations governing the large-scale dynamics of hard rods, including diffusive effects that go beyond ballistic transport. Diffusive corrections are the first-order terms in the hydrodynamic gradient expansion and we obtain them through an explicit microscopic calculation of the dynamics of hard rods. We show that they differ significantly from the prediction of Navier-Stokes hydrodynamics, as the correct hydrodynamics description is instead given by two coupled equations, giving respectively the evolution of the one point functions and of the connected two-point correlations. The resulting equations are time-reversible and reduce to the usual Navier-Stokes hydrodynamic equations in the limit of near-equilibrium evolution. This represents the first exact microscopic calculation showing how ballistic dynamics generates long-range correlations, in agreement with general results from the recently developed ballistic macroscopic fluctuation theory, and showing how such long range-correlations directly affect the diffusive hydrodynamic terms, in agreement with, and clarifying, recent related results.

# 1 Introduction

Understanding many-body dynamics is among the most challenging problems in contemporary physics. Real-world systems are typically large, composed of many interacting particles, and observed over time scales far exceeding the typical interaction time. While a wide array of numerical and analytical methods exists for studying few-body systems, large-scale systems out of equilibrium often remain beyond the reach of direct simulation.

An important theoretical tool in this context is the hydrodynamic approximation, which forgoes the description of individual particles in favor of densities of conserved quantities (e.g. particle number, momentum, and energy). These densities obey continuity equations whose simplest forms are the Euler and Navier–Stokes equations. Because these continuum equations are significantly simpler than the underlying microscopic dynamics, they provide a key starting point for analytical and numerical investigations.

Despite the empirical success of hydrodynamics, there is still no universally accepted derivation of these equations from a microscopic description. The standard approach relies on the principle of maximum entropy, which posits that, at each point in space, the system can be approximated by a thermal (local equilibrium) state. Even if this principle captures the general structure of the hydrodynamic equations, it relies on thermodynamic quantities that are model dependent and notoriously difficult to calculate. As a result, one cannot generally assess the accuracy of hydrodynamics for a specific microscopic model, echoing one of Hilbert's famous open problems.

In certain integrable one-dimensional models, an infinite number of conserved quantities allows for exact calculation of these thermodynamic coefficients. Such systems can be described by generalized hydrodynamics (GHD) [2,3] (see for instance the reviews [4–6]). One of the simplest nontrivial examples is the hard rods model, see for example [7–11], consisting

of classical rods (or spheres) of diameter $a$ in one dimension. Its Euler GHD equation reads

$$\partial_t \rho(t,x,p) + \partial_x\Big(v^{\text{eff}}(t,x,p)\rho(t,x,p)\Big) = 0, \tag{1}$$

where $\rho(t,x,p)$ is the density of particles (at macroscopic time $t$, macroscopic position $x$, and momentum $p$), and

$$v^{\text{eff}}(t,x,p) = \frac{p - a\int \mathrm{d}q\, q\, \rho(t,x,q)}{1 - a\int \mathrm{d}q\, \rho(t,x,q)} \tag{2}$$

is the effective velocity arising from particle–particle interactions. This equation was first derived in [12] and rigorously proven in [13]. Its diffusive extension—often referred to as the Navier–Stokes GHD—was obtained in [14]:

$$\partial_t \rho(t,x,p) + \partial_x\Big(v^{\text{eff}}(t,x,p)\rho(t,x,p)\Big) = \frac{1}{2\ell}\,\partial_x\left(\int \mathrm{d}q\, D(p,q)\,\partial_x\rho(t,x,q)\right), \tag{3}$$

where

$$D(p,q) = \frac{a^2}{1 - a\int \mathrm{d}q\, \rho(t,x,q)}\Big(\delta(p-q)\int \mathrm{d}q'\, \rho(t,x,q')|p-q'| - \rho(t,x,p)|p-q|\Big). \tag{4}$$

Here, $\ell \gg 1$ is the ratio between the macroscopic and microscopic scales.

In the Euler scaling limit ($\ell \to \infty$), the right-hand side vanishes, reducing the description to the Euler equation. However, in the large-$\ell$ region, the diffusive correction is physically important because it encodes entropy production, implying relaxation. In [14], the Navier–Stokes GHD equation was rigorously proven for short times $t \to 0^+$ for certain classes of initial local equilibrium states. The short-time validity is typically expected to extend to all times, as one assumes the system's state at all times to be correctly approximated by the process of local relaxation from local equilibrium states. One thus explains relaxation on hydrodynamic time scales by local entropy maximization.

One can incorporate integrability-breaking external potentials into the hard rods model; in such cases, the Navier–Stokes GHD framework predicts that thermal states survive as the unique long-time attractors. This scenario is confirmed numerically for many potentials, but a notable exception is the harmonic potential, where the system relaxes yet does not approach a thermal state [15,16]. This discrepancy, combined with the recent observations of anomalous fluctuations at diffusive scales in integrable systems [17–21], indicates a breakdown of the usual thermalization argument in harmonic traps [22,23], raising doubts about the general validity of the Navier–Stokes GHD approach.

In this work, we reexamine the Navier–Stokes GHD equation in the hard rods model and show that the standard derivation fails to remain valid beyond infinitesimally short (macroscopic) times. The crucial reason is the emergence of long-range correlations during the dynamics, as first established in [24,25]. These violate the assumption that at all times, the system can be described by local relaxation from local equilibrium states. Indeed, although averages of local observables, in the $\ell \to \infty$ limit, still tend to the values they take in local equilibrium states, as required for Euler-scale hydrodynamics, spatial correlations of local observables receive $1/\ell$ corrections. Such long-range, $1/\ell$ spatial correlations are simply absent in local equilibrium states, and not described by relaxation of local observables; yet they affect the diffusive scale in (3) (see also our companion paper [1]).

We derive, in the hard rods model, a new equation directly from the microscopic model. Unlike the standard Navier–Stokes GHD, it consists of a coupled system involving both the one-point function (the particle density) and the two-point correlation function. This is the specialisation to the hard rods model of the general equations proposed in our companion

paper [1]. This new theory is fully time-reversal invariant and therefore does not produce an intrinsic arrow of time. Consequently, it fails to describe thermalization as it is usually understood.

The remainder of this paper is organized as follows. In Section 2, we discuss the family of hard rods states under consideration, namely those with typical large deviation scaling. This assumption is natural in hydrodynamics and includes, for example, standard local equilibrium states. In Section 3, we outline the main steps of our microscopic derivation, with additional technical details provided in Appendix A. Finally, in Section 4, we compare our analytical results against extensive numerical simulations to confirm the new picture.

## 1.1 Summary of the main results

The main result of our paper is that the Navier-Stokes equation for the dynamics of hard rods must be corrected by an additional, non-trivial term compared to Eq (3). The additional term involves the long-range correlations: the two-point correlations of the density of quasiparticles at macroscopically small, but microscopically large, distances:

$$C(t,x,p,y,q) = \lim_{\ell \to \infty} \ell \, \langle \rho_e(t,x,p) \rho_e(t,y,q) \rangle^c, \tag{5}$$

$$C_{\text{LR}}(t,x,p,y,q) = \lim_{\epsilon \to 0^+} C(t,x,p,y,q) \theta(|x-y|-\epsilon). \tag{6}$$

The correlation function $C(t,x,p,y,q)$ is the Euler scaling limit [25, 26] of the equal-time correlation function of the empirical density (see Eq. (17) below) in the state $\langle \cdots \rangle$, which depends implicitly on the macroscopic variation length $\ell$, and the time $t$. The limit defining $C_{\text{LR}}(t,x,p,y,q)$ serves to omit the delta-function part of $C(t,x,p,y,q)$, which represent, in macroscopic coordinates and under Euler scaling, the correlations at microscopic distances.

For the quantity $\rho(t,x,p) = \langle \rho_e(t,x,p) \rangle$, we obtain the following hydrodynamic equation, up to, including, the diffusive scale $1/\ell$:

$$\partial_t \rho(t,x,p) = -\partial_x(v^{\text{eff}}(t,x,p)\rho(t,x,p)) + \frac{1}{2\ell}\partial_x\left[\int dq\, D(p,q)\partial_x\rho(t,x,q)\right] +$$

$$+ \frac{1}{\ell}\partial_x\left[\frac{a1^{\text{dr}}(t,x)}{(2\pi)^2}\int dq\,(q-p)C_{\text{LR}}^{\text{n}}(t,x-v^{\text{eff}}(t,x,p)0^+,p,x-v^{\text{eff}}(x,q)0^+,q)\right] + \mathcal{O}(1/\ell^2). \tag{7}$$

Here, $C^{\text{n}}(x_1,p_1,x_2,p_2)$ is the two-point correlation function in normal modes

$$C^{\text{n}}(t,x_1,p_1,x_2,p_2) = \frac{(2\pi)^2}{1^{\text{dr}}(t,x_1)1^{\text{dr}}(t,x_2)}$$

$$\int dq_1\, dq_2 \left[\delta(p_1-q_1) + a\frac{\rho(t,x_1,p_1)}{1^{\text{dr}}(x_1)}\right]\left[\delta(p_2-q_2) + a\frac{\rho(t,x_2,p_2)}{1^{\text{dr}}(x_2)}\right]C(t,x_1,q_1,x_2,q_2) \tag{8}$$

and we defined $1^{\text{dr}}(t,x) = 1 - a\int dq\,\rho(t,x,q)$. Notably, the equation can also be written as

$$\boxed{\partial_t\rho(t,x,p) + \partial_x(v^{\text{eff}}(t,x,p)\rho(t,x,p)) = -\frac{1}{\ell}\partial_x\left[\frac{a}{(2\pi)^2}\int dq\,\frac{p-q}{1^{\text{dr}}(t,x)}C_{\text{LR,sym}}^{\text{n}}(t,x,p,q)\right] + \mathcal{O}(1/\ell^2)}$$
$$\tag{9}$$

where $C_{\text{LR,sym}}^{\text{n}}(x,p,q)$ is the symmetric part of the two-point correlation function

$$C_{\text{LR,sym}}^{\text{n}}(t,x,p,q) = \frac{1}{2}\left(C^{\text{n}}(t,x-0^+,p,x,q) + C^{\text{n}}(t,x+0^+,p,x,q)\right). \tag{10}$$

Eq. (7) or equivalently Eq. (9) together with the well-known evolution equation (55) for the correlation function [26], form a closed set of two coupled equations.

As in [14], Eq. (7) is valid, starting from any slowly varying initial state (at time, say, $t = 0$), for the *forward macroscopic derivative* $\partial_t \rho(t, x, p) = \lim_{\epsilon \to 0^+} (\rho(t+\epsilon, x, p) - \rho(t, x, p))/\epsilon$ where the $1/\ell$ expansion is performed before the $\epsilon \to 0^+$ limit. It is obtained by a physical construction similar to that of [14]: the local current observables are evaluated by relaxation from the state on macroscopic time slice $t$, to macroscopic time $t + 0^+$. It is current observables at $t + 0^+$ that form the right-hand side. But the difference with [14] is that the state at time $t$ is not assumed to be in local equilibrium, i.e. locally described by a generalized Gibbs ensemble (GGE) and otherwise uncorrelated. Rather, long-range correlations are taken into account. This is important, as these are known to develop over time [24]. The quantity $C^n_{LR}(t, x - v^{eff}(x, p)0^+, p, x - v^{eff}(x, q)0^+, q)$ in the second term of order $1/\ell$ on the right-hand side of (7) is the correlation in the state on time slice $t$. Note that it is evaluated at small, $\propto 0^+$ macroscopic distances (after the limit $\ell \to \infty$ has been taken). That is,

$$C^n_{LR}(t, x - v^{eff}(x, p)0^+, p, x - v^{eff}(x, q)0^+, q) =$$
$$\lim_{\epsilon \to 0^+} \lim_{l \to \infty} \ell \left\langle \rho_e(t, x - v^{eff}(x, p)\epsilon, p) \rho_e(y - v^{eff}(y, q)\epsilon, q) \right\rangle^c. \quad (11)$$

This is how the result depends on the long-range correlations of the state at time $t$.

The simplified expression in (9) is valid for the left-hand side being the *instantaneous time derivative* instead of the forward macroscopic time derivative (see Subsection 1.3): in $\partial_t \rho(t, x, p) = \lim_{\epsilon \to 0^+} (\rho(t + \epsilon, x, p) - \rho(t, x, p))/\epsilon$ the $\epsilon \to 0^+$ limit is performed before the $1/\ell$ expansion. It reproduces the macroscopic derivative in (7) because no matter the initial state, the system relaxes immediately (i.e. faster than macroscopic time) to a "dynamically stable manifold", by developing appropriate long-range correlations such that the right-hand side of (9), on this stable manifold, amounts to the two $1/\ell$ terms on the right-hand side of (7) (see Subsection 1.2). Unlike (7), where the long-range correlations appear as an additional correction to Navier-Stokes GHD (3), in the alternative (9) the standard diffusion term is absent. This clearly indicates new physics behind the diffusive scale, outlined in our companion paper [1]. Currents simply take their Euler-scale form plus a term controlled by long-range correlations, giving Eq. (9) for instantaneous time derivatives of densities. The diffusive-scale corrections to the currents are therefore entirely obtained from the corrections these new long-range correlations give to averages of currents, in sharp contrast to the usual procedure.

Crucially, we find that *it is incorrect to obtain the currents at time $t + 0^+$, that determine the forward derivative of densities, by performing a relaxation process from some assumed local-equilibrium state at time $t$.*

The new equations (7), (9) differ in many ways from the previous (3). We highlight two important differences:

1. The diffusive dynamics requires a higher amount of information than in the Navier-Stokes equation, being intrinsically dependent on both one- and two- point functions in a system of two coupled equations. The solution to the Euler-scale hydrodynamic equation for conserved densities enters the evolution equation for the two-point correlation function, and its solution then enters the evolution equation for the diffusive scale hydrodynamic equation.

2. The new equations are symmetric under time-reversal (see section 3.5). Therefore, entropy cannot be always increasing. This is in stark contrast to (3), which is demonstrated to produce monotonously non-decreasing (and generically increasing) entropy.

## 1.2 Behavior for local equilibrium states and dynamically stable manifold

An important special case is when the system is in a local equilibrium state (say at time $t = 0$), where no long-range correlations are present, i.e. macroscopically separated regions are fully uncorrelated. Indeed, since there are no long-range correlations in a local equilibrium state, at $t = 0$ (7) correctly reduces to (3). This is an important consistency check, as in [14], the validity of Eq. (3) was rigorously proven for (a subfamily of) local equilibrium states.

Eq. (9) indicates correctly that instantaneous currents do not receive diffusive corrections in local equilibrium states: the right-hand side vanishes as $C^n_{\text{LR,sym}}(x, p, q) = 0$ in such states. However, this does not provide the forward macroscopic time derivative of densities. This is because local equilibrium states are unstable in time. To see this we can expand $C^n_{\text{LR}}(t, x, p, y, q)$ for short times (see Appendix D and Eq. (14) in [1]):

$$C^n_{\text{LR}}(t, x, p, y, q)\big|_{y \approx x, t \approx 0} = \tfrac{a}{2} 1^{\text{dr}}(x)\big[\partial_x \rho(x, p)\rho(x, q) - \rho(x, p)\partial_x \rho(x, q)\big] \times$$
$$\times \big[\text{sgn}(y - x) - \text{sgn}(y - x + (v^{\text{eff}}(x, p) - v^{\text{eff}}(y, q))t)\big] + \mathcal{O}(t^2). \quad (12)$$

We see that immediately (faster than macroscopic time) a finite jump of the long-range correlations at $x = y$ appears. We interpret this as a "relaxation" towards a "dynamically stable manifold" of states with non-trivial long-range correlations. In fact, for all times $t$ the long-range correlations of any state in the dynamically stable manifold have a jump, fixed by the local one-point function [1, 27]:

$$C_{\text{LR}}(t, x, p, y, q)\big|_{y \approx x} = \tfrac{a}{2} 1^{\text{dr}}(t, x)\big[\partial_x \rho(t, x, p)\rho(t, x, q) -$$
$$- \rho(t, x, p)\partial_x \rho(t, x, q)\big] \text{sgn}(y - x) + (\text{continuous}). \quad (13)$$

Here (continuous) represents a possible additive term, continuous at $x = y$.

The right-hand side of Eq. (9) must be evaluated in this "dynamically stable manifold" in order to obtain macroscopic variations[1]. The macroscopic forward derivative at time $t = 0$ is obtained if the right hand side of (9) is evaluated at $t = 0^+$ (i.e. after the relaxation to the "stable manifold"), see section 3.4 for more explanations.

## 1.3 Relation to derivation in [1]

The subtle difference between (7) and (9) highlights the difference in spirit between this work and our companion paper [1]: The general theory developed in [1] reduces to (9) for hard rods. This provides an independent check for the physical assumptions in [1].

In [1] the *instantaneous* current is computed: at $t = 0$, there is no diffusive correction to the current. However, this current is not useful for predicting the quasi-particle density after a short macroscopic time, because the diffusive correction of the current is discontinuous from $t = 0 \to 0^+$. In this work on the other hand we do not evaluate the instantaneous current, but instead take the *macroscopic forward time derivative* of the quasi-particle density. In this sense (7) can be seen as a forward time averaging of (9). This forward time averaging does not affect the equation unless at those special times, like $t = 0$, where the current is discontinuous. While (9) is always time-symmetric, the forward time averaging of (7) makes (7) formally break time reversal symmetry at these special times.

Note that despite these physical differences at these special times, the two PDE's (7) and (9) are both mathematically equivalent. They only differ on a measure zero set in time, hence their solutions are equal.

---

[1] We note that the term involving long-range correlations in (7) has the particularity that it is invariant under such sudden formation of long-range correlations from the initial state; although it is affected by their slow variations over time.

## 2   The hydrodynamic state

In hydrodynamics, the initial state is typically chosen as an average over an ensemble of initial configurations, described by a probability measure $\varrho$ over the phase-space. Natural states from the perspective of GHD are, for instance, local equilibrium states (in macroscopic coordinates):

$$\mathrm{d}\varrho = \mathbf{1}_{\Sigma_N}(x_1,\ldots,x_N)e^{-\sum_i \beta(x_i,p_i)}\,\mathrm{d}^N x\,\mathrm{d}^N p\,. \tag{14}$$

Here $\mathbf{1}_{\Sigma_N}(x_1,\ldots,x_N)$ excludes unphysical states from the measure, being $\mathbf{1}_{\Sigma_N}(x_1,\ldots,x_N) = 1$ if no two hard rods overlap, and $\mathbf{1}_{\Sigma_N}(x_1,\ldots,x_N) = 0$ if at least two overlap:

$$\mathbf{1}_{\Sigma_N}(x_1,\ldots,x_N) = \begin{cases} 1 & (|x_i - x_j| > a/\ell \ \forall \ i \neq j) \\ 0 & (\text{otherwise}). \end{cases} \tag{15}$$

If $\beta(x,p) = \beta(p)$ does not depend on $x$, this describes an stationary state (a generalised Gibbs ensemble, GGE) with generalized inverse temperatures encoded within the function $\beta(p)$. On the other hand, if $\beta(x,p)$ is space dependent, the state is a non-equilibrium state on the macroscopic scale $\ell$, while still behaving as a GGE on microscopic scales. The central object of GHD is the quasi-particle density

$$\rho(x,p) = \langle \rho_{\mathrm{e}}(x,p) \rangle\,, \tag{16}$$

which is the average of the empirical density of particles

$$\rho_{\mathrm{e}}(x,p) = \tfrac{1}{\ell}\sum_i \delta(x-x_i)\delta(p-p_i) \tag{17}$$

over this probability measure $\varrho$, denoted by $\langle\ldots\rangle$. While $\rho_{\mathrm{e}}(x,p)$ is a 'spiky' function, after averaging over (14) it becomes a smooth function as $\ell \to \infty$. On the Euler scale (i.e. including only $\mathcal{O}\big(1/\ell^0\big)$ terms), the GHD equation is a closed equation describing the evolution of the quasi-particle density.

As a matter of fact, a basic assumption behind hydrodynamics states that the connected correlation functions (also called cumulants) of $\rho_{\mathrm{e}}(x,p)$ show large deviation scaling:

$$\langle \rho_{\mathrm{e}}(x_1,p_1)\ldots\rho_{\mathrm{e}}(x_k,p_k) \rangle^{\mathrm{c}} \sim 1/\ell^{k-1}\,. \tag{18}$$

For instance, this scaling is satisfied by the states (14). For such a reason, the Euler scale equation (1) only depend on $\rho(x,p)$; all higher connected correlation functions vanish as $\ell \to \infty$.

**Assumption 1** *We assume that the state satisfies the large deviation scaling as $\ell \to \infty$, i.e. (18).*

### 2.1   The connected two-point correlation function

In this paper we aim to derive the diffusive equation including $\mathcal{O}(1/\ell)$ terms and, hence, to describe the state using the first two connected correlation functions $\rho(x,p)$ and $\langle \rho_{\mathrm{e}}(x,p)\rho_{\mathrm{e}}(y,q)\rangle^{\mathrm{c}}$, being all higher order correlations negligible.

Firstly, it is important to observe that the two point connected correlation function has the following general form:

$$C(x,p,y,q) = \ell\,\langle \rho_{\mathrm{e}}(x,p)\rho_{\mathrm{e}}(y,q)\rangle^{\mathrm{c}} = \delta(x-y)C_{\mathrm{GGE}}(x,p,q) + C_{\mathrm{LR}}(x,p,y,q) + \mathcal{O}(1/\ell), \tag{19}$$

where

$$C_{\mathrm{GGE}}(x,p,q) = \rho(x,p)\delta(p-q) + \rho(x,p)\rho(x,q)\left(-2a + a^2\int \mathrm{d}q'\,\rho(x,q')\right) \tag{20}$$

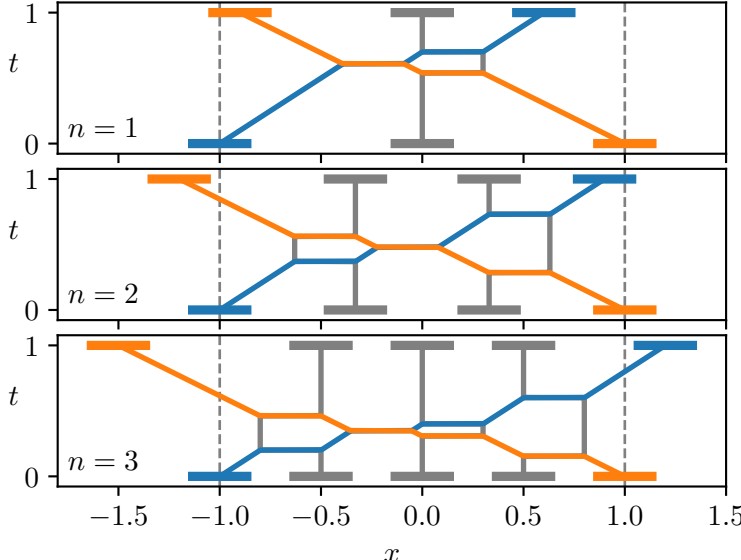

Figure 1: Explanation of long range correlations: As time evolves (see (21)) two hard rods (in this case the blue and the orange one), become correlated because they interact with the same particles between them. If the number of particles between them is lower, then they will travel less far, if they are higher, both will travel further. Since the density of particles in a region fluctuates around its mean value, this means that particles which travel through the same region become correlated. In the hydrodynamic limit, this can then be observed as non-trivial long-range correlations of the density.

are the correlations of the GGE that corresponds to the local quasi-particle density $\rho(x, p)$. $C_{\text{LR}}(x, p, y, q)$ are long range correlations in the system, which are piecewise smooth functions that can present jumps [1, 27]. In the typical scenario, the initial state only present non-long range correlations (which is the case in local equilibrium states (14)), while the long range correlations are developed during time-evolution. In Fig. 1, we give a pictorial representation of the mechanism behind the creation of long range correlations.

**Assumption 2** *We assume that the connected two-point correlation function is of the form* (19). *In particular, this is satisfied for all times t if the evolution starts from a local equilibrium state* (14)*, whose connected two-point correlations function evolution can be computed explicitly (see Appendix D).*

While the delta part $C_{\text{GGE}}(x, p, q)$ follows the evolution of $\rho(x, p)$, being a constant functional of particles density, the long range contribution $C_{\text{LR}}(x, p, y, q)$ will evolve non-trivially [1, 24, 25, 27] (we give the full evolution formula in Appendix D). Since hard rods are described by a local theory, only the behavior at $x \approx y$ can affect the diffusive equation. Interestingly, the long range correlations have a jump, given by (13), at this point (see Fig.2) [1, 27].

Let us remark that, microscopically, for $x/\ell \approx y/\ell$ the correlations are always finite, but present a complicated exponentially fast decay [28] (see Appendix F). Instead, at macroscopic scales, after the coarse graining they emerge as the $\delta(x - y)$ contribution. However, as apparent from our derivation, the hydrodynamic theory on diffusive scale does not need any information about the microscopical structure of two point correlations, except for its symmetry under exchange of $x$ and $y$ (following from PT symmetry).

**Assumption 3** *The microscopic shape giving rise to the singular $\delta(x - y)$ term in* (19) *is microscopically symmetric in $x - y$ as $\ell \to \infty$.*

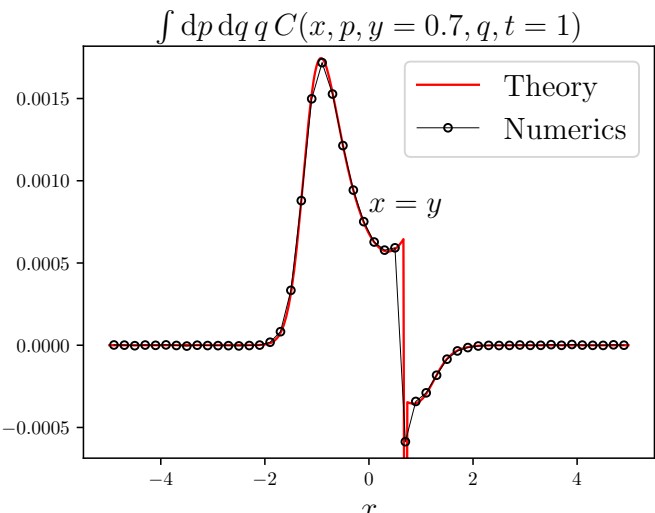

Figure 2: Integrated correlations $\int \mathrm{d}p\,\mathrm{d}q\,qC(x,y,p,q,t)$ at $y = 0.7$ and $t = 1$, as function of $x$. The initial state is defined in Eq. (60). The black circles represent the numerical simulation (which are done as described in section 4), while the red curve the theoretical prediction, see Appendix D. This figure shows how the system developed long range correlations during the dynamics and that they are perfectly captured by the theory. Also, at $x = y$, we observe a discontinuity. For numerical simulation we use the same data as in [1].

**Remark 1** *We chose the singular part of the correlations to be given by the GGE correlations since this is a natural physical choice (it is satisfied for instance by the local GGE states (14)). However, in principle our derivation can be straight-forwardly adapted to initial states, where the singular part is given by other local correlation functions. In this case the resulting diffusive GHD equation will be different.*

## 3 Derivation of the diffusive equation

The aim of this paper is to derive the emergent evolution equation for hard rods in the Euler scaling limit $\ell \to \infty$ (described more detailed below) which will be correct up to diffusive order, i.e. up to including $\mathcal{O}(1/\ell)$.

For clarity, let us consider temporarily space-time coordinates to be on microscopic scale. As a matter of fact, the solution to the microscopic system is explicitly known in hard rods [12]: Given the initial (microscopic) positions $x_i$ and momenta $p_i$ of $N$ hard rods with length $a$, the trajectory of a (tracer) particle $i$ is given by

$$x_i(t) = x_i + p_i t + a \sum_{j \neq i} \theta(\hat{x}_i + p_i t - \hat{x}_j - p_j t) - \theta(\hat{x}_i - \hat{x}_j), \tag{21}$$

where the so called contracted coordinates are

$$\hat{x}_i = x_i - a \sum_{j \neq i} \theta(x_i - x_j). \tag{22}$$

**Remark 2** *Physical hard rods exchange their momenta during collisions. However, it is often convenient to additionally exchange both particles (this is merely a relabeling). This, way particles*

*keep their momenta during scattering, but exchange their positions. Equation* (21) *describes the so called tracer dynamics.*

**Remark 3** *The intuitive idea behind* (21) *is as follows (see for instance [12, 29]): The contracted coordinates $\hat{x}_i$ represents the physically available space, i.e. the left most particle keeps its position, the first particle gets shifted by $-a$, the second one by $-2a$ and so on. In these $\hat{x}$ coordinates the evolution is free, i.e. $\hat{x}_i(t) = \hat{x}_i + p_i t$. After evolving to time $t$, one has to order the particles again in $\hat{x}$ space and undo the contraction, i.e. the left most particle remains put, the first one gets shifted by $+a$, the second one by $+2a$ and so on. This is implemented in* (21).

We now specify the Euler scaling limit: we are interested in situations where the particle number $N$, the length-scale of observation $\ell$ and the time-scale of observation $T$ are sent to infinity, while keeping their ratios fixed $N \sim \ell \sim T \to \infty$. First, let us rescale time and space to macroscopic quantities $x_i \to x_i \ell$, $\hat{x}_i \to \hat{x}_i \ell$ and $t \to t\ell$. In terms of rescaled variables, Eq. (21) and (22) read:

$$x_i(t) = x_i + p_i t + \frac{a}{\ell} \sum_{j \neq i} \theta(\hat{x}_i + p_i t - \hat{x}_j - p_j t) - \theta(\hat{x}_i - \hat{x}_j), \tag{23}$$

$$\hat{x}_i = x_i - \frac{a}{\ell} \sum_{j \neq i} \theta(x_i - x_j). \tag{24}$$

The latter effectively amounts to rescaling the rod size $a \to a/\ell$. From now on we will always consider macroscopic coordinates.

## 3.1 Derivation of the time-evolution

We now consider the evolved particles positions (23) and average it over the initial state. In particular we aim to express it in terms of quasi-particle density:

$$\rho(t, x, p) = \langle \rho_e(t, x, p) \rangle = \left\langle \frac{1}{\ell} \sum_i \delta(x - x_i(t)) \delta(p - p_i) \right\rangle. \tag{25}$$

Also, it is convenient to integrate it against a test function $\phi(x, p)$:

$$\int \mathrm{d}x\, \mathrm{d}p\, \rho(t, x, p) \phi(x, p) = \left\langle \frac{1}{\ell} \sum_i \phi(x_i(t), p_i) \right\rangle = \left\langle \frac{N}{\ell} \phi(x_1(t), p_1) \right\rangle$$

$$= \int \mathrm{d}x_1\, \mathrm{d}p_1\, \rho(x_1, p_1) \langle \phi(X_e(t, x_1, p_1), p_1) | x_1, p_1 \rangle. \tag{26}$$

Here $\langle \ldots | x_1, p_1 \rangle$ is the conditional expectation value given known values of $x_1$ and $p_1$, and

$$X_e(t, x_1, p_1) = x_1 + p_1 t + \frac{a}{\ell} \sum_{j \neq 1} \theta(\hat{x}_1 + p_1 t - \hat{x}_j - p_j t) - \theta(\hat{x}_1 - \hat{x}_j) \tag{27}$$

is the trajectory of a particle starting at $x_1, p_1$. Since $\phi(x, p)$ is a smooth function in $x$ we can expand around $x = \langle X_e(t, x_1, p_1) | x_1, p_1 \rangle$:

$$\int \mathrm{d}x\, \mathrm{d}p\, \rho(t, x, p) \phi(x, p) = \int \mathrm{d}x_1\, \mathrm{d}p_1 \Big[ \rho(x_1, p_1) \phi(\langle X_e(t, x_1, p_1) | x_1, p_1 \rangle, p_1) +$$

$$+ \frac{1}{2} \rho(x_1, p_1) \partial_x^2 \phi(\langle X_e(t, x_1, p_1) | x_1, p_1 \rangle, p_1) \mathrm{Var}[X_e(t, x_1, p_1) | x_1, p_1] \Big] + \mathcal{O}(1/\ell^2). \tag{28}$$

Here we used the assumption on the scaling of higher-order correlation functions of the particle density to neglect all terms beyond the variance

$$\mathrm{Var}[X_e(t, x_1, p_1)|x_1, p_1] = \left\langle (X_e(t, x_1, p_1) - \langle X_e(t, x_1, p_1)|x_1, p_1 \rangle)^2|x_1, p_1 \right\rangle \sim 1/\ell. \tag{29}$$

Next, let us expand $\langle X_e(t, x_1, p_1)|x_1, p_1 \rangle$ in $1/\ell$:

$$\langle X_e(t, x_1, p_1)|x_1, p_1 \rangle = X(t, x_1, p_1) + \tfrac{1}{\ell}\Delta X(t, x_1, p_1) + \mathcal{O}\left(1/\ell^2\right). \tag{30}$$

Inserting this into (28) we have:

$$\begin{aligned}
\int \mathrm{d}x\,\mathrm{d}p\,\rho(t, x, p)\phi(x, p) =\ & \int \mathrm{d}x_1\,\mathrm{d}p_1\,\rho(x_1, p_1)\phi(X(t, x_1, p_1), p_1) \\
& + \tfrac{1}{\ell}\int \mathrm{d}x_1\,\mathrm{d}p_1\,\rho(x_1, p_1)\partial_x\phi(X(t, x_1, p_1), p_1)\Delta X(t, x_1, p_1) \\
& + \tfrac{1}{2}\int \mathrm{d}x_1\,\mathrm{d}p_1\,\rho(x_1, p_1)\partial_x^2\phi(X(t, x_1, p_1), p_1)\mathrm{Var}[X_e(t, x_1, p_1)|x_1, p_1] + \mathcal{O}\left(1/\ell^2\right).
\end{aligned} \tag{31}$$

Here $X(t, x, p)$ is the Euler-scale GHD characteristic. Indeed, on the Euler scale particles follow deterministic trajectories. When going to diffusive scale there are two more effects that need to be taken into account. First, due to the fluctuations in the initial state the distribution of a particle at time $t$ becomes a Gaussian with width $\sqrt{\mathrm{Var}[X_e(t, x_1, p_1)|x_1, p_1]} \sim 1/\sqrt{\ell}$. Second, there is also a deterministic shift of order $1/\ell$ to the mean value of the Gaussian, described by $\Delta X(t, x_1, p_1)$. If both $\Delta X(t, x_1, p_1)$ and $\mathrm{Var}[X_e(t, x_1, p_1)|x_1, p_1]$ are known for all $x_1$ and $p_1$ it fully describes the particle trajectories on the diffusive scale and in turn also determine $\rho(t, x, p)$. Indeed, denoting $X^{-1}(t, x, p)$ the inverse function to $X(t, x, p)$ and removing the test function from Eq. (31), we find

$$\begin{aligned}
\rho(t, x, p) = \rho_E(t, x, p) &- \tfrac{1}{\ell}\partial_x(\rho(t, x, p)\Delta X(t, X^{-1}(t, x, p), p)) \\
&+ \tfrac{1}{2\ell}\partial_x^2(\rho(t, x, p)V(t, X^{-1}(t, x, p), p)) + \mathcal{O}\left(1/\ell^2\right), \quad (32)
\end{aligned}$$

where we defined the Euler scale evolved quasi-particle density

$$\rho_E(t, x, p) = \rho(X^{-1}(t, x, p), p)\frac{\mathrm{d}X^{-1}(t, x, p)}{\mathrm{d}x}, \tag{33}$$

and where we used $\mathrm{Var}[X_e(t, x_1, p_1)|x_1, p_1] = \tfrac{1}{\ell}V(t, x_1, p_1) + \mathcal{O}\left(1/\ell^2\right)$.

In Appendix A we give the formulas for $X(t, x_1, p_1)$, $\Delta X(t, x, p)$ and $V(t, x, p)$, see (A.14), (A.15) and (A.16). Note that the formula for the Euler scale trajectory

$$X(t, x_1, p_1) = x_1 + p_1 t + a\int \mathrm{d}x_2\,\mathrm{d}p_2\,\rho(x_2, p_2)(\theta(\hat{X}(x_1) - \hat{X}(x_2) + (p_1 - p_2)t) - \theta(x_1 - x_2)), \tag{34}$$

where

$$\hat{X}(x) = x - a\int_{-\infty}^{x} \mathrm{d}y\,\mathrm{d}q\,\rho(y), \tag{35}$$

is simply the continuous limit of (23).

## 3.2 Obtaining the diffusive equation

Eq. (32) gives the quasi-particle distribution $\rho(t,x,p)$ on the diffusive scale for an arbitrary time $t$. In the following, we derive the diffusive PDE having Eq. (32) as solution.

In order to derive an evolution equation from the solution (32), we need to study the behavior at time $t \to 0^+$. First, let us note that:

$$\lim_{t\to 0^+} X(t,x,p) = x, \qquad \lim_{t\to 0^+} \Delta X(t,x,p) = 0, \qquad \lim_{t\to 0^+} V(t,x,p) = 0, \qquad (36)$$

which are derived in appendix A. This implies that

$$\lim_{t\to 0} \rho(t,x,p) = \rho(x,p) = \rho(0,x,p) + \mathcal{O}(1/\ell^2). \qquad (37)$$

This statement might seem trivial, but it is important to stress that it is not. In fact, it is trivial only at a microscopic time, i.e. taking $t \to 0^+$ before $\ell \to \infty$. However, in our derivation, we first send $\ell \to \infty$ while keeping $t$ on finite Euler scale and then we consider $t \to 0^+$. This way we can probe the system only at long microscopic times. A failure of (36) would physically mean that the system locally equilibrates before any Euler-scale time. The fact (36) is true, is ultimately connected to the fact that we choose a state with local GGE correlations as initial state. In fact, even if our derivation can formally be applied to initial states with non-equilibrium local correlations, it would bring immediate local equilibration before any Euler scale evolution happens. As a conclusion, at $t = 0^+$ the state is expected to present local equilibrium correlations, independently from the precise shape at $t = 0$. As a consequence, we can always consider initial states that satisfies (36).

Let us now take the time derivative of (32) to obtain:

$$\lim_{t\to 0^+} \partial_t \rho(t,x,p) = -\partial_x(\partial_t X(0^+,x,p)\rho(x,p)) - \tfrac{1}{\ell}\partial_x(\rho(x,p)\partial_t \Delta X(0^+,x,p))$$
$$+ \tfrac{1}{2\ell}\partial_x^2(\rho(x,p)\partial_t V(0^+,x,p)) + \mathcal{O}(1/\ell^2), \quad (38)$$

where $v^{\mathrm{eff}}(x,p) = \lim_{t\to 0^+} \partial_t X(t,x,p)$ is the effective velocity of a GHD characteristic.

This equation finally gives the evolution equation of $\rho(t,x,p)$ at time $t = 0$. However, because the structure of the state remains invariant under time and the hard rods dynamics do not have any memory, this immediately implies that (38) holds at all times.

Taking the time derivative of (34) and sending $t \to 0^+$, we recover the well-known formula for the effective velocity

$$\partial_t X(0^+,x,p) = v^{\mathrm{eff}}(x,p) = \frac{p - d\int dq\, q\rho(x,q)}{1 - d\int dq\, \rho(x,q)}. \qquad (39)$$

The expressions for the $1/\ell$ correction terms are more complicated and are derived in Appendix A. In general we can split them into two parts, one stems from the singular GGE part of the correlations (20) and the other from the long range part of the correlations:

$$\partial_t \Delta X(0^+,x,p) = \partial_t \Delta X_{\mathrm{local}}(0^+,x,p) + \partial_t \Delta X_{\mathrm{LR}}(0^+,x,p), \qquad (40)$$
$$\partial_t V(0^+,x,p) = \partial_t V_{\mathrm{local}}(0^+,x,p) + \partial_t V_{\mathrm{LR}}(0^+,x,p). \qquad (41)$$

The contributions from the local GGE correlations are

$$\partial_t \Delta X_{\mathrm{local}}(0^+,x,p) = a^2 \int dq \left( \partial_x \rho(x,p) + \frac{a}{2\, 1^{\mathrm{dr}(x)}}\rho(x,q) \int dq'\, \partial_x \rho(x,q') \right) \frac{|p-q|}{1^{\mathrm{dr}(x)}}, \qquad (42)$$

$$\partial_t V_{\mathrm{local}}(0^+,x,p) = a^2 \int dq\, \rho(x,q) \frac{|p-q|}{1^{\mathrm{dr}(x)}}, \qquad (43)$$

where $1^{\text{dr}}(x) = 1 - d \int dq\, \rho(x,q)$. The long range correlations give rise to:

$$\partial_t \Delta X_{\text{LR}}(0^+, x, p) = a \int dq\, \frac{p-q}{1^{\text{dr}}(x)\rho(x,p)} \int dp'\, dq' \left[\delta(p'-p) + a\frac{\rho(x,p)}{1^{\text{dr}}(x)}\right] \times$$
$$\times \left[\delta(q'-q) + a\frac{\rho(x,q)}{1^{\text{dr}}(x)}\right] C_{\text{LR}}(x - v^{\text{eff}}(x,p)0^+, p', x - v^{\text{eff}}(x,q)0^+, q') \qquad (44)$$
$$= \frac{a1^{\text{dr}}(x)}{(2\pi)^2} \int dq\, \frac{p-q}{\rho(x,p)} C^{\text{n}}_{\text{LR}}(x - v^{\text{eff}}(x,p)0^+, p, x - v^{\text{eff}}(x,q)0^+, q),$$

$$\partial_t V_{\text{LR}}(0^+, x, p) = 0. \qquad (45)$$

In (44) we introduced the correlations of $n(x,p) = 2\pi\rho(x,p)/1^{\text{dr}}$ (see Appendix A (A.32)):

$$C^{\text{n}}(x, p, y, q) = \langle n(x,p)n(y,q)\rangle^{\text{c}} = \delta(x-y)C^{\text{n}}(x,p,q) + C^{\text{n}}_{\text{LR}}(x,p,y,q). \qquad (46)$$

Note the point splitting implied by the $0^+$, which determines the choice of side at the jump of $C_{\text{LR}}(x, p, y, q)$ at $x = y$. The physical meaning is simple: For finite $t$ the correlations are evaluated at the origin of the GHD characteristics, which for $t \to 0$ is approximately given by $x - v^{\text{eff}}(x,p)t$.

We can insert these expressions into (38) and find the following diffusive equation:

$$\partial_t \rho(x,p) = -\partial_x(v^{\text{eff}}(x,p)\rho(x,p)) + \frac{1}{2\ell}\partial_x\left[\int dq\, D(p,q)\partial_x\rho(x,q)\right] +$$
$$+ \frac{1}{\ell}\partial_x\left[\frac{a1^{\text{dr}}(x)}{(2\pi)^2}\int dq\,(q-p)C^{\text{n}}_{\text{LR}}(x - v^{\text{eff}}(x,p)0^+, p, x - v^{\text{eff}}(x,q)0^+, q)\right]. \qquad (47)$$

In the last expression (47), the second term in RHS comes from the local GGE correlations and coincides, as expected, with the usual diffusion matrix in a GGE state:

$$D(p,q) = \frac{a^2}{1^{\text{dr}}(x)}\left(\delta(p-q)\int dq'\,\rho(x,q')|p-q'| - \rho(x,p)|p-q|\right). \qquad (48)$$

The other term is the novel contribution which comes from the long range correlations. It is important to observe that the point-splitting in the correlation function shows how to treat the discontinuity in the correlation function, see (13). Since the long range correlations present a jump at $x = y$, we can split them locally into a symmetric and an antisymmetric part:

$$C^{\text{n}}_{\text{LR}}(x,p,y,q)\big|_{x\approx y} = C^{\text{n}}_{\text{LR,sym}}(x,p,q) + \text{sgn}(y-x)C^{\text{n}}_{\text{LR,asym}}(x,p,q)$$
$$= \frac{1}{2}\left(C^{\text{n}}_{\text{LR}}(x-0^+,p,x,q) + C^{\text{n}}_{\text{LR}}(x+0^+,p,x,q)\right) \qquad (49)$$
$$+ \frac{1}{2}\text{sgn}(y-x)\left(C^{\text{n}}_{\text{LR}}(x-0^+,p,x,q) - C^{\text{n}}_{\text{LR}}(x+0^+,p,x,q)\right).$$

As explicitly derived in Appendix D, the jump has the following form:

$$C^{\text{n}}_{\text{LR,asym}}(x,p,q) = \frac{a}{21^{\text{dr}}(x)^2}(2\pi)^2[\partial_x\rho(x,p)\rho(x,q) - \partial_x\rho(x,q)\rho(x,p)]. \qquad (50)$$

Inserting this formula into Eq.(7), we find that this produces a term that exactly cancels the original Kubo diffusion (48). Therefore the diffusive GHD equation can be also written as:

$$\partial_t \rho(x,p) = -\partial_x(v^{\text{eff}}(x,p)\rho(x,p)) + \frac{1}{\ell}\partial_x\left[\frac{a}{(2\pi)^2}\int dq\, \frac{q-p}{1^{\text{dr}}(x)}C^{\text{n}}_{\text{LR,sym}}(x,p,q)\right]. \qquad (51)$$

This is the main result of this paper. In this form the diffusion does not depend on any singular parts of the correlations anymore, only on the continuous part of the long range correlations. Note that we already know that the solution to this equation is given by (32).

**Remark 4** *The fact that the jump contribution exactly cancels the contribution from the singular GGE correlations is quite interesting. The reason for this cancellation is more evident in our more heuristic and more general derivation [1]: There the fact that the singular GGE correlations and the jump do not affect the diffusive dynamics is due to fluid-cell averaging. From the perspective of fluid-cell averaging only the current on the boundary of the fluid-cell is important for the dynamics of the total charge inside the fluid-cell. However, the jump and the GGE correlations are fully contained inside a fluid-cell, hence they should not be able to affect the dynamics.*

### 3.3   Evolution equation of correlation functions

We see that the equation for $\rho(x,p)$ requires the knowledge of the correlations at time $t$. It is well-known that these satisfy the linearized Euler equations in both components [26]

$$\partial_t \langle \rho_{\rm e}(t,x,p)\rho_{\rm e}(s,y,q)\rangle^{\rm c} + \partial_x \left[ \int \mathrm{d}k \, \frac{\delta j(t,x,p)}{\delta \rho(k)} \langle \rho_{\rm e}(t,x,k)\rho_{\rm e}(s,y,q)\rangle^{\rm c} \right] = 0 \qquad (52)$$

$$\partial_s \langle \rho_{\rm e}(t,x,p)\rho_{\rm e}(s,y,q)\rangle^{\rm c} + \partial_y \left[ \int \mathrm{d}k \, \frac{\delta j(s,y,q)}{\delta \rho(k)} \langle \rho_{\rm e}(t,x,p)\rho_{\rm e}(s,y,k)\rangle^{\rm c} \right] = 0, \qquad (53)$$

where $j(x,p) = v^{\rm eff}(x,p)\rho(x,p)$. In particular, the so called flux Jacobian can be explicitly expressed as

$$\frac{\delta j(x,p)}{\delta \rho(k)} = v^{\rm eff}(x,p)\delta(p-k) - \frac{ak\rho(x,p)}{1^{\rm dr}(x)} + a\frac{v^{\rm eff}(x,p)\rho(x,p)}{1^{\rm dr}(x)} . \qquad (54)$$

Hence, we find the equation

$$\partial_t C(x,p,y,q) + \partial_x \left[ \int \mathrm{d}k \, \frac{\delta j(t,x,p)}{\delta \rho(k)} C(x,k,y,q) \right] + + \partial_y \left[ \int \mathrm{d}k \, \frac{\delta j(t,y,q)}{\delta \rho(k)} C(x,p,y,k)^{\rm c} \right] = 0. \qquad (55)$$

Here we see that (51) and (55) are a closed set of equations, which both together form the diffusive GHD equations.

Alternatively the correlation functions can also be computed explicitly at time $t$, see Appendix D.

### 3.4   Relation to Boldrighini-Suhov results

Boldrighini and Suhov, in [14], derived a Navier-Stokes diffusive equation for the hard rods model with the following expression:

$$\partial_t \rho(x,p) = -\partial_x(v^{\rm eff}(x,p)\rho(x,p)) + \tfrac{1}{2\ell}\partial_x \left[ \int \mathrm{d}q \, D(p,q)\partial_x \rho(x,q) \right]. \qquad (56)$$

However, as motivated in this paper and in [1], *their result only applies to states with a constant particle density* $\int \mathrm{d}q\,\rho(x,q)$ *and no long range correlations.* For such states, the last part of (7) vanishes, due to the absence of long range correlations, and only the usual Kubo-type diffusion (48) exists. This formula then therefore agrees with (56).

However, evolving such a state for any small $t > 0$, long range correlations will be developed. Therefore, at any time $t = 0^+$, Eq. (51) is expected to predict the correct time evolution of the state. And indeed this also gives rise to the same formula as (56). We derive in appendix D that in this case

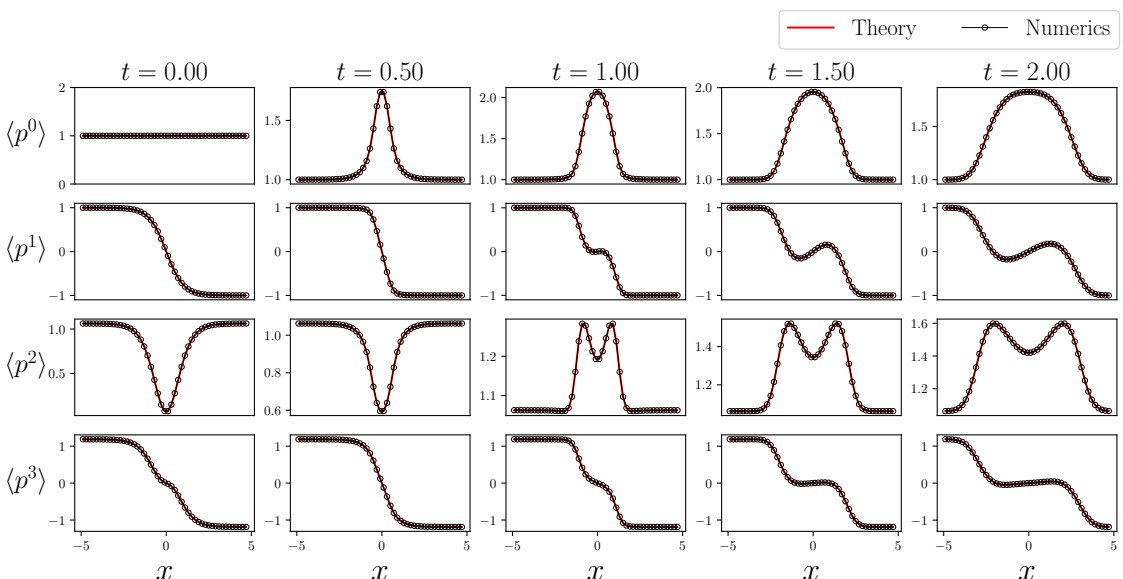

Figure 3: Evolution of $n$-moments $\langle p^n \rangle(x,t)$ of hard rods velocity distribution from the initial state (60) as a function of space, for $n \in \{0,1,2,3\}$ and for $t \in \{0,0.5,1,1,5,2\}$. Empty circles represent the numerical simulation of the hard rods gas with $\ell = 200$, while red lines represent the theoretical prediction of Eq. (22). The numerical data are averaged over an ensemble of $3 \times 10^6$ different realizations.

$$
\begin{aligned}
C_{\text{LR,sym}}^{\text{n}}(t=0^+,x,p,q) &= \\
&= \frac{a(2\pi)^2}{2 1^{\text{dr}}(t,x)^2} \, \text{sgn}(p-q)[\rho(0,x,p)\partial_x\rho(0,x,q) - \rho(0,x,q)\partial_x\rho(0,x,p)],
\end{aligned} \quad (57)
$$

which after inserting into (51) indeed reproduces (56).

Note that this shows that the local equilibrium correlations, where no long range correlations are present, are unstable from the perspective of GHD. At any other Euler time $t > 0$ the long range correlations appear instantly. This shape has to build up on time-scales much smaller than the Euler-scale. This can be seen as a local 'equilibration', not to a local equilibrium state, but rather to a stable fully out of equilibrium state (the jump of the correlations violates PT symmetry, thus making it an out-of-equilibrium state).

## 3.5 Time-reversibility

In (3) the entropy

$$
S[\rho] = \int \mathrm{d}x \, \mathrm{d}p \, \rho(x,p) \log \tfrac{\rho(x,p)}{1^{\text{dr}}(x)}, \quad (58)
$$

is always non-decreasing in time. This means that there is an 'arrow of time', which allows for the distinction between forward and backward time evolution (entropy increase is forward time evolution, entropy decrease is backward time evolution)

However, the new diffusive GHD equations (51) and (55) are fully time-reversibly: Time-reversal symmetry means $t \to -t$ and $p \to -p$, which further implies

$$
\rho(x,p) \to \rho(x,-p), \quad v^{\text{eff}}(x,p) \to -v^{\text{eff}}(x,p), \quad C^{\text{n}}(x,p;y,q) \to C^{\text{n}}(x,-p;y,-q). \quad (59)
$$

It is easy to see that with these replacements (51) and (55) remain invariant. The time-reversal symmetry also prevents the existence of an always non-decreasing entropy and thus, in these equations there is no 'arrow of time'.

The intuitive reason for this is as follows: entropy increase is associated to loss of information. In (3) the implicit assumption is that all information except for the one-point function immediately is lost due to thermalization. Therefore, time evolution is not reversible. In the correct theory (51) and (55), on the other hand we need to consider both the one-point and two-point function. This way all relevant information about this system is available at all times and hence the time-evolution is reversible.

## 4   Numerical simulation

In this section we compare the exact solution derived in Sec. 3 with numerical simulations of the hard rods dynamics. In particular, we consider an initial local GGE state (3) determined by the following space varying particle density

$$\rho(x, p, t = 0) = \exp\left(-(p + \tanh(x))^2/2\sigma\right)/\sqrt{2\pi}\sigma, \tag{60}$$

where, as in the previous sections, $x$ and $t$ represent macroscopic variables. The related microscopic variables are $x^{\text{micro}} \equiv x\ell$, $t^{\text{micro}} \equiv t\ell$. The state (60) represents a smoothened version of the sharp partition protocol, where the average particle velocity changes from $-1$ at $x \to \infty$ to 1 at $x \to -\infty$. Also, the averaged density of particles is initially homogeneous, as $\int \mathrm{d}p\, \rho(x, p, t = 0) \equiv \bar{\rho}(x) = 1$. In particular, in a local GGE state with constant particle density the hard rods' positions do not depend on the momenta and are given by a Poisson point process in the volume-excluded coordinates $\hat{x}_i = x_i - ai$ [14]. Hence, we can generate initial particle positions as $x_{i+1}^{\text{micro}} = x_i^{\text{micro}} + a + (1/\bar{\rho} - a)\xi_i$, where $\xi_i$ are i.i.d. standard exponentially distributed variable. The momenta are then chosen randomly according the to the local rapidity distribution (60). The time evolution of this system can be trivially performed when mapped to free particles coordinates $\hat{x}$, i.e. using Eq. (22). Indeed, for each initial state, we follow the following algorithm: we map the hard rods coordinate to free particles coordinates $\{x_i \to \hat{x}_i\}$; we evolve the free particles coordinates to the final time $t_f$ using $\{\hat{x}_i(t_f) = \hat{x}_i(t_0) + p_i t_f\}$; we use the inverse mapping (23) to evaluate the hard rods position at final time.

In Figure 3 we show the space dependent $n$-moments of hard rods velocity distribution

$$\langle p^n \rangle(x, t) = \int \mathrm{d}p\, p^n \rho(x, p, t), \tag{61}$$

with $n \in \{0, 1, 2, , 3\}$, for different times $t \in \{0, 0.5, 1, 1.5, 2\}$ and using macroscopic scale $\ell = 200$. The ensemble average is performed over $3 \times 10^6$ realizations. The system is initialized on the total region $x \in [-20, 20]$, in order to avoid boundary effects on the measured interval $x \in [-5, 5]$ for $t \leq 2$. The empty dots, representing the results of numerical simulations, are compared with the theoretical predictions given by the Euler evolution obtained from Eq.(34) (red solid lines). As expected, the theoretical prediction match perfectly with the numerical results.

As a matter of facts, diffusive effects appear at order $O(1/\ell)$ and hence are not qualitatively visible from Fig. 3. In Figure 4 we quantitatively evaluate the $O(1/\ell)$ correction to Euler dynamics of $\langle p^n \rangle(x, t)$, here called $\Delta \langle p^n \rangle_{1/\ell}$. More precisely, we simulate the system for different macroscopic lengths $\ell \in \{100, 120, 140, 160, 180, 200\}$ and we perform a fit of $\langle p^n \rangle(x, t; \ell)$ with model $f(\ell) = f_1 + f_2/\ell$, for each point in space and time. Hence, we estimate $\Delta \langle p^n \rangle_{1/\ell}$ as the fitted parameter $f_2(x, t)$, represented by the black line. The associated

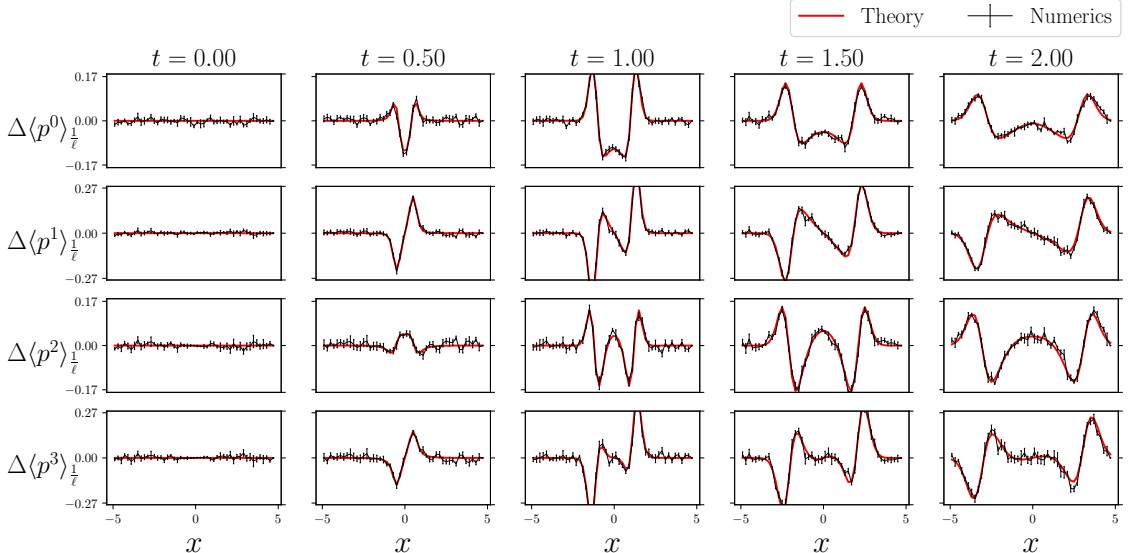

Figure 4: Time evolution of the $O(1/\ell)$ correction $\Delta\langle p^n\rangle_{1/\ell}$ to the $n$-moments of hard rods velocity distribution from the initial state (60) as a function of space, for $n \in \{0,1,2,3\}$ and for $t \in \{0,0.5,1,1,5,2\}$. The black line represent the numerical simulation of the hard rods gas, evaluated as the $f_2$ parameter of a fit with model $f(\ell) = f_1 + f_2/\ell$ to $\langle p^n\rangle(x,t;\ell)$, for $\ell \in \{100,120,140,160,180,200\}$. The fit is performed independently for each point in space and time and the associated error bar is the standard deviation of the fit parameter. The numerical data are averaged over an ensemble of $3\times 10^6$ different realizations. The red lines represent the theoretical prediction of Eq. (31).

error is determined as the standard deviation of the fit parameter. The red solid lines represent the theoretical prediction given by Eq. (31) (red solid lines). The theoretical results are in perfect agreement with numerical simulations at all times and for all moment of particle distribution, with discrepancies being always below the error bars.

## 4.1 Checking the differential equation

So far we checked the validity of the explicit 'solution' to (51). However, we are actually interested in checking the equation (51) and comparing to (3). We can easily initialize the system in a local equilibrium state (14), but we know that in this case (51) reduces to (3). So we need to compare in a different state: a state that is physical but also contains long-range correlations. How can we create such a state? The simple idea to use time-evolution: We initialize the system in the local equilibrium state (60) and then evolve it for $t = 1$. Since long-range correlations develop during the hydrodynamic evolution, the state at $t = 1$ will contain long-range correlations. Then we numerically evaluate the $O(1/\ell)$ correction to the time derivative $\Delta[\partial_t\langle p^n\rangle]_{1/\ell}$ at time $t = 1$. More precisely, we estimate it through the difference quotient

$$\partial_t\langle p^n\rangle(x,t;\ell) \simeq \frac{\langle p^n\rangle(x,t+\Delta t;\ell) - \langle p^n\rangle(x,t-\Delta t;\ell)}{2\Delta t} \tag{62}$$

for $\ell \in \{500,600,\ldots,1000\}$ and using $\Delta t = 0.05$. The average is taken over an ensemble of $8 \times 10^{10}$ initial states. Then, $\Delta[\partial_t\langle p^n\rangle]_{1/\ell}$ is computed using the same fit as for Fig. 4 and, again, the error bars are estimated as the standard deviation of the fit parameter. The red dots represent the prediction from Eq. (51), while green dots are predictions from Navier-Stokes GHD (3). Finally, blue circles represent the numerical time derivative of the solution (32),

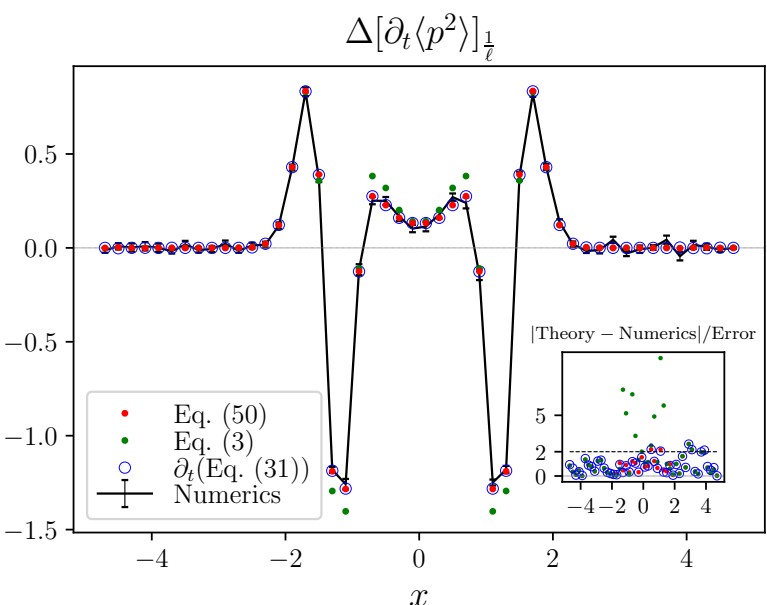

Figure 5: We show the $O(1/\ell)$ correction to time derivative of the second moment of hard rods velocity distribution from the initial state (60), as a function of space at $t = 1$. The black line represent the numerical simulation of the hard rods gas, for which the time derivative is evaluated through the difference quotient (62) with $\Delta t = 0.05$. Then, the $O(1/\ell)$ correction is estimated as the $f_2$ parameter of a fit with model $f(\ell) = f_1 + f_2/\ell$ to $\partial_t \langle p^2 \rangle(x, t; \ell)$, for $\ell \in \{500, 600, \ldots, 1000\}$. The fit is performed independently for each point in space and the associated error bar is the standard deviation of the fit parameter. The numerical data are averaged over an ensemble of $8 \times 10^{10}$ different realizations. The red dots represent the theoretical prediction of the new theory Eq. (51), while green dots are the prediction due to the Navier-Stokes-like theory Eq. (3). Finally, blue circles represent the time derivative of the solution (32), computed through difference quotient with $\Delta t = 0.01$. The inset shows the distance between theoretical predictions and data points in error bars units. One can clearly see that the new theory fits the numerical simulations much better, while the old Navier-Stokes-like theory Eq. (3) differs significantly. For numerical simulation we use the same data as in [1].

computed through difference quotient with $\Delta t = 0.01$. The inset shows the distance between theory and numerics, normalized with the error bars. As expected, both the theories are in good agreement with the numerical simulations. But, meanwhile Eq. (3) shows deviations much bigger than the standard deviation in few points, Eq. (51) is in perfect agreement for any observed $x$ presenting differences always smaller than two times the error bars.

## 5   Conclusion

In this manuscript we derived the equation that governs the diffusive correction to the Euler (generalized) hydrodynamic description of the integrable hard rods model. The starting point for this is the exact microscopic solution formula (23). Assuming the usual thermodynamic/hydrodynamic large deviation scaling of correlation functions, we perform the large-scale $\ell \to \infty$ limit of (23), keeping the Euler $\sim 1/\ell^0$ and the diffusive contributions $\sim 1/\ell$.

Then, taking a $t \to 0^+$ limit we finally obtain the equation governing the diffusive dynamics. We find that, unlike the previous Navier-Stokes like equation (3) [14], the equation for the one-point function up to, including, the diffusive scale, depends on the two-point function at the Euler scale. The equation for the Euler-scale two-point function depends on the Euler-scale one-point function, but not on higher order correlation functions (those can be neglected due to the assumed large deviation scaling). Therefore, the diffusive dynamics are described by a closed system of two coupled equations, instead of one, which can be solved in a "graded" way: Euler-scale one-point function → Euler-scale two-point function → diffusive scale one-point function. The equation reduces to the Navier-Stokes like equation (3) if and only if correlations are of local equilibrium type; this is important as this was rigorously proven in [14]. However, in time additional long range correlations emerge, scaling like $1/\ell$. Therefore, they do not affect the Euler dynamics, but are relevant on the diffusive scale. Interestingly, these long range correlations have a jump at $x = y$, whose contribution to the diffusive equation exactly cancels the normal Navier-Stokes term. Therefore, the remaining parts of the long-range correlations are the only drive for diffusive dynamics. Due to this cancellation the diffusive equation is time-reversible. Therefore, unlike the previous theory (3), entropy cannot be always increasing.

Similar equations like (51) can also be derived more generally based on general hydrodynamic principles for models with linearly degenerate hydrodynamics (which includes integrable systems), see our companion paper [1]. Therefore, the microscopic derivation presented in this paper provides an important independent verification of the derivation in [1]. Hard rods are a special model in this regard, as it has an explicit microscopic solution available. If there is another model with an available explicit solution, a similar derivation will be possible. It would also be interesting to extend the derivation presented here into a rigorous mathematical proof.

Another interesting direction is to extend this derivation to higher order beyond the diffusive scale, for instance to derive the equation governing dispersive hydrodynamics. The current theory is given by [30]. Since this is based on similar assumptions as (3) it seems reasonable to expect that dispersive hydrodynamics is different as well. We expect in general that the $n$-point correlation functions at hydrodynamic order $k$ (Euler is $k = 0$, diffusive is $k = 1$, etc.), is governed by a dynamical equation involving $m$-point correlation functions at orders $l$ with $(m, l) \neq (m, k)$, and $0 \leq l \leq k$ and $1 \leq m \leq n + k - l$, thus giving a hierarchical set of closed equations.

Such a hierarchical system of equations for the evolution of few point functions is reminiscent of the celebrated BBGKY theory describing the evolution of generic interacting many body systems. Crucially, our theory presents some remarkable differences with respect to BBGKY. First, being a hydrodynamic theory, the dynamical $n$-point functions are defined on coarse grained time and space variables, while BBGKY is defined microscopically. Second, our theory admits states with non trivial correlations (see Eq. (20)), while BBGKY typically assumes uncorrelated initial states in order to define a proper truncation scheme. Finally, while in BBGKY the hierarchy is truncated perturbatively in the interaction potential strength, here a truncation scheme is automatically given by the hydrodynamic scale $1/\ell$, withouth assumption on the strength of interaction potentials.

Finally, since our new theory is time-reversal invariant it cannot describe thermalization to a GGE. Previously, it was thought that this was due to diffusion as suggested by the entropy increase of (3). Therefore, thermalization to a GGE has to happen in a different way, which is not yet clear. As a related problem, one can also study how quickly an integrable model thermalizes to a Gibbs state in an integrability breaking external potential. This requires to first extending the diffusive formula to situations including an external potential [15,31]. In particular, the new diffusive formula might help to understand why hard rods do not to thermalize

in a harmonic external potential [15,22,23]. In the new theory, the evolution is affected by the two-point correlations. Therefore, it might be possible that non-trivial correlations stabilize non-thermal equilibria.

## Acknowledgements

We thank Herbert Spohn for useful discussions. Numerical computations were partially done in Julia [32], in particular using the ApproxFun.jl library [33], and ran on the CREATE cluster [34].

**Funding information**   FH is funded by the faculty of Natural, Mathematical & Engineering Sciences at King's College London. BD was supported by the Engineering and Physical Sciences Research Council (EPSRC) under grants EP/W010194/1 and EP/Z534304/1. J.D.N. and L.B. are funded by the ERC Starting Grant 101042293 (HEPIQ) and the ANR-22-CPJ1-0021-01.

## A   Details of the derivation

As explained in the main text, in order to compute the exact asymptotic expectation value of the quasi-particle density $\rho$ up to diffusive order, we need to compute two pieces of information: For a particle starting at $z_1 = (x_1, p_1)$ we need to know its average position and variance

$$\langle X_{\mathrm{e}}(t, z_1) | z_1 \rangle = X(t, z_1) + \tfrac{1}{\ell} \Delta X(t, z_1) + \mathcal{O}\big(1/\ell^2\big), \tag{A.1}$$

$$\mathrm{Var}[X_{\mathrm{e}}(t, z_1) | z_1] = \tfrac{1}{\ell} V(t, z_1) + \mathcal{O}\big(1/\ell^2\big) \tag{A.2}$$

after time $t$. This computation is lengthy, but actually straight-forward: It consists of averaging the exact expression (27) over the initial state. We will do so by using standard techniques from large deviation theory, which will allow us to write explicit expressions only in terms of the first two cumulants

$$\rho(0, x, p) = \langle \rho_{\mathrm{e}}(0, x, p) \rangle, \tag{A.3}$$

$$C(x, p; y, q) = \langle \rho_{\mathrm{e}}(0, x, p) \rho_{\mathrm{e}}(0, y, q) \rangle^{\mathrm{c}} =$$
$$= \langle \rho_{\mathrm{e}}(0, x, p) \rho_{\mathrm{e}}(0, y, q) \rangle - \langle \rho_{\mathrm{e}}(0, x, p) \rangle \langle \rho_{\mathrm{e}}(0, y, q) \rangle \tag{A.4}$$

of the initial state. For convenience we will restrict the initial correlations to be of the form (19), but the procedure can easily be extended to more general initial correlations.

First, let us define the two-particle marginal distribution

$$f_2(x_1, p_1; x_2, p_2) = \left\langle \frac{1}{\ell^2} \sum_{i \neq j} \delta(x - x_i) \delta(p - p_i) \delta(y - x_j) \delta(q - p_j) \right\rangle \tag{A.5}$$

and write

$$\langle X_{\mathrm{e}}(t, z_1) | z_1 \rangle = x_1 + p_1 t + a \int \mathrm{d}x_2 \, \mathrm{d}p_2 \, \frac{f_2(z_1; z_2)}{\rho(z_1)} \langle \theta(\hat{x}_1 + p_1 t - \hat{x}_2 - p_2 t) - \theta(\hat{x}_1 - \hat{x}_2) | z_1; z_2 \rangle. \tag{A.6}$$

Throughout the derivation we will use the following simple argument from large deviation theory: Assume that $Y$ satisfies a large deviation principle, i.e. the $n$'th cumulant scales as

$\ell^{-(n-1)}$. Then we can compute the expectation value of any smooth function $\psi(y)$ by the following *cumulants expansion*:

$$
\begin{aligned}
\langle \psi(Y) \rangle &= \left\langle \psi(\langle Y \rangle) + \psi'(\langle Y \rangle)(Y - \langle Y \rangle) + \tfrac{1}{2}\psi''(\langle Y \rangle)(Y - \langle Y \rangle)^2 + \tfrac{1}{6}\psi'''(\langle Y \rangle)(Y - \langle Y \rangle)^3 + \dots \right\rangle \\
&= \psi(\langle Y \rangle) + \tfrac{1}{2}\psi''(\langle Y \rangle)\mathrm{Var}[Y] + \mathcal{O}\big(1/\ell^2\big).
\end{aligned}
\tag{A.7}
$$

Applying this idea on (A.6) we find

$$
\langle X_e(t,z_1)|z_1\rangle = x_1 + p_1 t + a\int \mathrm{d}x_2\, \mathrm{d}p_2\, \frac{f_2(z_1;z_2)}{\rho(z_1)} \theta(\langle \hat{x}_1 - \hat{x}_2|z_1;z_2\rangle + (p_1 - p_2)t) - \theta(x_1,x_2) +
$$

$$
+ \frac{a}{2}\int \mathrm{d}x_2\, \mathrm{d}p_2\, \frac{f_2(z_1;z_2)}{\rho(z_1)}\delta'(\langle \hat{x}_1 - \hat{x}_2|z_1;z_2\rangle + (p_1 - p_2)t)\mathrm{Var}[\hat{x}_1 - \hat{x}_2|z_1;z_2]. \tag{A.8}
$$

Of course, technically $\theta(x)$ is not a differentiable function, but (A.7) can still be applied in a distributional sense (i.e. integrating both sides against a test function). Note that

$$
\hat{x}_2 - \hat{x}_1 = x_2 - x_1 - \frac{a}{\ell}\,\mathrm{sgn}(x_2 - x_1) - a n_{(x_1,x_2)}, \tag{A.9}
$$

where $n_{(x_1,x_2)} = N_{(x_1,x_2)}/\ell$ counts the number of particles between $x_1 < x_i < x_2$ (we define $n_{(x_1,x_2)} = -n_{(x_2,x_1)}$ for $x_2 < x_1$). This is a thermodynamic quantity and will thus satisfy a large-deviation principle. Its expectation value and variance are computed in Appendix B:

$$
\langle n_{(x_1,x_2)}|z_1;z_2\rangle = \int_{x_1}^{x_2} \mathrm{d}z_3\, \rho(z_3) + \frac{1}{\ell}\sum_{i=1,2}\int_{x_1}^{x_2} \mathrm{d}z_3\, \frac{f_2^c(z_i;z_3)}{\rho(z_i)} + \mathcal{O}\big(1/\ell^2\big), \tag{A.10}
$$

$$
\mathrm{Var}[n_{(x_1,x_2)}|z_1;z_2] = \mathrm{Var}[n_{(x_1,x_2)}] + \mathcal{O}\big(1/\ell^2\big) \tag{A.11}
$$

$$
= \int_{x_1}^{x_2} \mathrm{d}z_3\, \mathrm{d}z_4\, f_2^c(z_3;z_4) + \frac{1}{\ell}\,\mathrm{sgn}(x_2 - x_1)\int_{x_1}^{x_2} \mathrm{d}z_3\, \rho(z_3) + \mathcal{O}\big(1/\ell^2\big). \tag{A.12}
$$

Let us define $\hat{X}(x) = x - a\int_{-\infty}^{x} \mathrm{d}y\, \mathrm{d}q\, \rho(y,q)$ and write:

$$
\langle X_e(t,z_1)|z_1\rangle = x_1 + p_1 t + a\int \mathrm{d}z_2\, \rho(z_2)(\theta(\hat{X}(x_1) - \hat{X}(x_2) + (p_1 - p_2)t) - \theta(x_1 - x_2)) +
$$

$$
+ \frac{a^2}{\ell}\int \mathrm{d}z_2\, \rho(z_2)\delta(\hat{X}(x_1) - \hat{X}(x_2) + (p_1 - p_2)t)\left[\sum_{i=1,2}\int_{x_1}^{x_2} \mathrm{d}z_3\, \frac{f_2^c(z_i;z_3)}{\rho(z_i)} + \mathrm{sgn}(x_2 - x_1)\right] +
$$

$$
+ \frac{a^3}{2}\int \mathrm{d}z_2\, \rho(z_2)\delta'(\hat{X}(x_1) - \hat{X}(x_2) + (p_1 - p_2)t)\mathrm{Var}[n_{(x_1,x_2)}] +
$$

$$
+ a\int \mathrm{d}x_2\, \mathrm{d}p_2\, \frac{f_2^c(z_1;z_2)}{\rho(z_1)}(\theta(\hat{X}(x_1) - \hat{X}(x_2) + (p_1 - p_2)t) - \theta(x_1 - x_2)) + \mathcal{O}\big(1/\ell^2\big)
$$

$$
\tag{A.13}
$$

Here we defined $f_2^c(z_1;z_2) = f_2(z_1;z_2) - \rho(z_1)\rho(z_2) = \mathcal{O}(1/\ell)$ and discarded all terms $\mathcal{O}\big(1/\ell^2\big)$. Simplifying the $\delta$ functions we finally find $\langle X_e(t,z_1)|z_1\rangle = X(t,z_1) + \frac{1}{\ell}\Delta X(t,z_1) + \mathcal{O}\big(1/\ell^2\big)$ with:

$$
X(t,z_1) = x_1 + p_1 t + a\int \mathrm{d}z_2\, \rho(z_2)(\theta(\hat{X}(x_1) - \hat{X}(x_2) + (p_1 - p_2)t) - \theta(x_1 - x_2)), \tag{A.14}
$$

$$\Delta X(t,z_1) = +a \int dx_2 \, dp_2 \, \frac{Lf_2^c(z_1;z_2)}{\rho(z_1)}(\theta(\hat{X}(x_1) - \hat{X}(x_2) + (p_1 - p_2)t) - \theta(x_1 - x_2)) +$$

$$+ \frac{a^3}{2} \int dp_2 \, \frac{1}{1^{dr}(x_2)} \partial_{x_2} \Big[ \frac{\rho(x_2, p_2)}{1^{dr}(x_2)} \ell \text{Var}[n_{(x_1, x_2)}] \Big] \Big|_{x_2 = X(\hat{X}(x_1) + (p_1 - p_2)t)} + \tag{A.15}$$

$$+ a^2 \int dp_2 \, \frac{\rho(x_2, p_2)}{1^{dr}(x_2)} \Bigg[ \sum_{i=1,2} \int_{x_1}^{x_2} dz_3 \, \frac{f_2^c(z_i; z_3)}{\rho(z_i)} + \text{sgn}(x_2 - x_1) \Bigg] \Bigg|_{x_2 = X(\hat{X}(x_1) + (p_1 - p_2)t)}.$$

Using similar steps one can also compute:

$$\ell \text{Var}[X_e(t, z_1)|z_1] = 2a^3 \int dz_2 \, dp_3 \, (\theta(\hat{X}(x_1) + p_1 t - \hat{X}_2(x_2) - p_2 t) - \theta(x_1 - x_2)) \times$$

$$\times \rho(z_2) \frac{\rho(z_3)}{1^{dr}(x_3)} \Bigg( \int_{x_1}^{x_3} dz \, \frac{f_2^c(z_2; z)}{\rho(z_2)} + \mathbf{1}_{(x_1, x_3)}(x_2) \Bigg) \Bigg|_{x_3 = X(\hat{X}(x_1) + (p_1 - p_3)t)} +$$

$$+ a^4 \int dp_2 \, dp_3 \, \frac{\rho(z_2)}{1^{dr}(x_2)} \frac{\rho(z_3)}{1^{dr}(x_3)} \text{Cov}[n_{(x_1, x_2)}, n_{(x_1, x_3)}] \Bigg|_{x_i = X(\hat{X}(x_1) + (p_1 - p_i)t)} + \tag{A.16}$$

$$+ a^2 \int dz_2 \, dz_3 \, f_2^c(z_2; z_3)(\theta(\hat{X}(x_1) + p_1 t - \hat{X}_2(x_2) - p_2 t) - \theta(x_1 - x_2)) \times$$

$$\times (\theta(\hat{X}(x_1) + p_1 t - \hat{X}(x_3) - p_3 t) - \theta(x_1 - x_3)) +$$

$$+ a^2 \int dz_2 \, \rho(z_2)(\theta(\hat{X}(x_1) + p_1 t - \hat{X}_2(x_2) - p_2 t) - \theta(x_1 - x_2))^2 + \mathcal{O}(1/\ell).$$

Here $\text{Cov}[n_{(x_1, x_2)}, n_{(x_1, x_3)}]$ is given by (B.13).

Into these expressions we now need to insert the initial correlations. Since (A.15) and (A.16) are linear in $f_2^c$, we can treat the local and long-range part of the correlations separately.

## A.1 Local GGE initial correlations

We start from an initial state which has GGE correlations, i.e.

$$f_2^c(z_1; z_2) = \frac{1}{\ell} \delta(x_1 - x_2) \gamma(x_1, p_1, p_2), \tag{A.17}$$

where

$$\gamma(x_1, p_1, p_2) = \rho(x_1, p_1) \rho(x_2, p_2)(-2a + a^2 \bar{\rho}(x_1)), \tag{A.18}$$

with $\bar{\rho}(x) = \int dp \, \rho(x, p)$. Note that:

$$\int dp_2 \, \gamma(x_1, p_1, p_2) = \rho(x_1, p_1)(1^{dr}(x_1)^2 - 1), \tag{A.19}$$

$$\int_{x_1}^{x_2} dx_3 \, dp_3 \, f_2^c(z_3; z_4) = \rho(z_4)(1^{dr}(x_4)^2 - 1)\mathbf{1}_{(x_1, x_2)}(x_4). \tag{A.20}$$

We also need to evaluate the last expression in the case when $z_4 = z_1, z_2$ in which case the indicator function is undefined. Thus, at this point one has to investigate the microscopic structure of $f_2^c(z_3, z_1)$ close to $x_3 = x_1$. Conveniently, this microscopic structure is symmetric, which implies that the correct regularization is $1/2$:

$$\int_{x_1}^{x_2} dx_3 \, dp_3 \, f_2^c(z_3; z_1) = \frac{1}{2} \rho(z_1)(1^{dr}(x_1)^2 - 1) \text{sgn}(x_2 - x_1), \tag{A.21}$$

$$\int_{x_1}^{x_2} dx_3 \, dp_3 \, f_2^c(z_3; z_2) = \frac{1}{2} \rho(z_2)(1^{dr}(x_2)^2 - 1) \text{sgn}(x_2 - x_1). \tag{A.22}$$

Let us define $\Gamma(x) = \int_{-\infty}^{x} dx\, \bar{\rho}(x) 1^{dr}(x)^2$ and simplify (A.12) and (B.13):

$$\text{Var}[n_{(x_1,x_2)}] = \frac{1}{\ell} \int_{x_1}^{x_2} dx_3 \, d\, dp_4 \, \gamma(x_3, p_3, p_4) + \frac{1}{\ell} \text{sgn}(x_2 - x_1) \int_{x_1}^{x_2} dx_3 \, \bar{\rho}(x_3)$$

$$= \frac{1}{\ell}(\Gamma(x_1 \vee x_2) - \Gamma(x_1 \wedge x_2)), \quad \text{(A.23)}$$

$$\text{Cov}[n_{(x_1,x_2)}, n_{(x_1,x_3)}] = \frac{1}{\ell}\Big(\theta(x_2 - x_1)\theta(x_3 - x_1)(\Gamma(x_2 \wedge x_3) - \Gamma(x_1))$$

$$+ \theta(x_1 - x_2)\theta(x_1 - x_3)(\Gamma(x_1) - \Gamma(x_2 \vee x_3))\Big). \quad \text{(A.24)}$$

Now we can evaluate $\Delta X_{\text{local}}(t, x, p)$ from (A.15):

$$\Delta X_{\text{local}}(t, x_1, p_1) = a \int dp_2 \frac{\gamma(x_1, p_1, p_2)}{\rho(x_1, p_1)}(\theta((p_1 - p_2)t) - \frac{1}{2}) +$$

$$+ \frac{a^3}{2} \int dp_2 \frac{1}{1^{dr}(x_2)} \partial_{x_2}\Big[\frac{\rho(x_2, p_2)}{1^{dr}(x_2)}(\Gamma(x_1 \vee x_2) - \Gamma(x_1 \wedge x_2))\Big]\Big|_{x_2 = X(\hat{X}(x_1) + (p_1 - p_2)t)} +$$

$$+ \frac{a^2}{2} \int dp_2 \frac{\rho(x_2, p_2)}{1^{dr}(x_2)} \text{sgn}(x_2 - x_1)\Big[1^{dr}(x_1)^2 + 1^{dr}(x_2)^2\Big]\Big|_{x_2 = X(\hat{X}(x_1) + (p_1 - p_2)t)}$$

$$= \frac{a}{2} \int dp_2 \frac{\gamma(x_1, p_1, p_2)}{\rho(x_1, p_1)} \text{sgn}(p_1 - p_2) + \quad \text{(A.25)}$$

$$+ \frac{a^3}{2} \int dp_2 \frac{1}{1^{dr}(x_2)} \partial_{x_2}\Big[\frac{\rho(x_2, p_2)}{1^{dr}(x_2)}(\Gamma(x_1 \vee x_2) - \Gamma(x_1 \wedge x_2))\Big]\Big|_{x_2 = X(\hat{X}(x_1) + (p_1 - p_2)t)} +$$

$$+ \frac{a^2}{2} \int dp_2 \frac{\rho(x_2, p_2)}{1^{dr}(x_2)} \text{sgn}(p_1 - p_2)\Big[1^{dr}(x_1)^2 + 1^{dr}(x_2)^2\Big]\Big|_{x_2 = X(\hat{X}(x_1) + (p_1 - p_2)t)},$$

and $V_{\text{local}}(t, x_1, p_1)$ from (A.16)

$$V_{\text{local}}(t, x_1, p_1) = 2a^3 \int dz_2 \, dp_3 \, (\theta(\hat{X}(x_1) + p_1 t - \hat{X}_2(x_2) - p_2 t) - \theta(x_1 - x_2)) \times$$

$$\times \rho(z_2)\frac{\rho(z_3)}{1^{dr}(x_3)} \mathbf{1}_{(x_1, x_3)}(x_2)\bar{\rho}(x_2)1^{dr}(x_2)^2\Big|_{x_3 = X(\hat{X}(x_1) + (p_1 - p_3)t)} +$$

$$+ a^4 \int dp_2 \, dp_3 \frac{\rho(z_2)}{1^{dr}(x_2)}\frac{\rho(z_3)}{1^{dr}(x_3)} \ell \, \text{Cov}[n_{(x_1, x_2)}, n_{(x_1, x_3)}]\Big|_{x_i = X(\hat{X}(x_1) + (p_1 - p_i)t)} + \quad \text{(A.26)}$$

$$+ a^2 \int dx_2 \, dp_2 \, dp_3 \, \gamma(x_2, p_2, p_3)(\theta(\hat{X}(x_1) + p_1 t - \hat{X}_2(x_2) - p_2 t) - \theta(x_1 - x_2)) \times$$

$$\times (\theta(\hat{X}(x_1) + p_1 t - \hat{X}(x_2) - p_3 t) - \theta(x_1 - x_2)) +$$

$$+ a^2 \int dz_2 \, \rho(z_2)(\theta(\hat{X}(x_1) + p_1 t - \hat{X}_2(x_2) - p_2 t) - \theta(x_1 - x_2))^2.$$

In the limit $t \to 0$ these formulas become (36), (42) and (43).

## A.2 Long range correlations

Now let us study the additional contribution that arises in the presence of long range correlations. They are described by a non-singular $f_2^c(x_1, p_1, x_2, p_2)$, however, we will allow it to have a jump at $x_1 = x_2$:

$$f_2^c(x_1, p_1, x_2, p_2) = \theta(x_2 - x_1)\beta_+(x_1, p_1, x_2, p_2) + \theta(x_1 - x_2)\beta_-(x_1, p_1, x_2, p_2) =$$
$$= \beta_{\mathrm{sgn}(x_2 - x_1)}(x_1, p_1, x_2, p_2). \quad \text{(A.27)}$$

Both $\beta_+$ and $\beta_-$ are assumed to be smooth at least in some neighborhood around $x_1 = x_2$. Inserting this into (A.15) and (A.16) we directly obtain a well-defined formula. In the limit $t \to 0$, it is easy to see that

$$\Delta X_{\mathrm{LR}}(t, x_1, p_1)\Big|_{t \to 0^+} = V_{\mathrm{LR}}(t, x_1, p_1)\Big|_{t \to 0^+} = \frac{\mathrm{d}}{\mathrm{d}t} V_{\mathrm{LR}}(t, x_1, p_1)\Big|_{t \to 0^+} = 0. \quad \text{(A.28)}$$

The only non-trivial contribution to the diffusive equation is:

$$\frac{\mathrm{d}}{\mathrm{d}t}\Delta X_{\mathrm{LR}}(t, x_1, p_1)\Big|_{t \to 0^+} =$$
$$= a^2 \int \mathrm{d}p_2 \frac{\rho(x_1, p_2)}{1^{\mathrm{dr}}(x_2)} \left( \int \mathrm{d}p \frac{\beta_{\mathrm{sgn}(p_1 - p_2)}(x_1, p_1, x_1, p)}{\rho(x_1, p_1)} + \frac{\beta_{\mathrm{sgn}(p_1 - p_2)}(x_1, p, x_1, p_2)}{\rho(x_1, p_2)} \right) \frac{p_1 - p_2}{1^{\mathrm{dr}}(x_1)}$$
$$+ \frac{a^3}{2} \int \mathrm{d}p_2 \frac{\rho(x_1, p_2)}{1^{\mathrm{dr}}(x_1)} \left( \int \mathrm{d}p\, \mathrm{d}p'\, \beta_+(x_1, p, x_1, p') + \beta_-(x_1, p, x_1, p') \right) \frac{p_1 - p_2}{1^{\mathrm{dr}}(x_1)}$$
$$+ a \int \mathrm{d}p_2 \frac{\beta_{\mathrm{sgn}(p_1 - p_2)}}{\rho(x_1, p_1)} \frac{p_1 - p_2}{1^{\mathrm{dr}}(x_1)}. \quad \text{(A.29)}$$

This can be written in compact form as follows:

$$\rho(x_1, p_1)\frac{\mathrm{d}}{\mathrm{d}t}\Delta X_{\mathrm{LR}}(t, x_1, p_1)\Big|_{t \to 0^+} =$$
$$= a \int \mathrm{d}p_2 \frac{p_1 - p_2}{1^{\mathrm{dr}}(x_1)} \int \mathrm{d}p\, \mathrm{d}p' \left[ \delta(p - p_1) + a\frac{\rho(x_1, p_1)}{1^{\mathrm{dr}}(x_1)} \right] \times$$
$$\times \left[ \delta(p' - p_2) + a\frac{\rho(x_1, p_2)}{1^{\mathrm{dr}}(x_1)} \right] \beta_{\mathrm{sgn}(p_1 - p_2)}(x_1, p, x_1, p'). \quad \text{(A.30)}$$

We can compare this expression to the expression for the correlation functions of the occupation function $n(x, p) = 2\pi\frac{\rho(x, p)}{1^{\mathrm{dr}}(x)}$, which responses to a small perturbation $\delta\rho(x, p)$ as

$$\delta n(x, p) = 2\pi\frac{\delta\rho(x, p)}{1^{\mathrm{dr}}(x)} + 2\pi\frac{\rho(x, p)}{1^{\mathrm{dr}}(x)^2}a\int \mathrm{d}q\, \delta\rho(x, q) =$$
$$= \frac{2\pi}{1^{\mathrm{dr}}(x)}\int \mathrm{d}q\left[ \delta(p - q) + a\frac{\rho(x, p)}{1^{\mathrm{dr}}(x)} \right]\delta\rho(x, q) \quad \text{(A.31)}$$

and therefore:

$$C^{\mathrm{n}}(x_1, p_1, x_2, p_2) = \langle \delta n(x_1, p_1)\delta n(x_2, p_2) \rangle =$$
$$= \frac{(2\pi)^2}{1^{\mathrm{dr}}(x_1)1^{\mathrm{dr}}(x_2)}\int \mathrm{d}q_1\, \mathrm{d}q_2 \left[ \delta(p_1 - q_1) + a\frac{\rho(x_1, p_1)}{1^{\mathrm{dr}}(x_1)} \right] \times$$
$$\times \left[ \delta(p_2 - q_2) + a\frac{\rho(x_2, p_2)}{1^{\mathrm{dr}}(x_2)} \right] C(x_1, q_1, x_2, q_2). \quad \text{(A.32)}$$

Comparing with (A.30) we can identify:

$$\rho(x_1, p_1)\frac{\mathrm{d}}{\mathrm{d}t}\Delta X_{\mathrm{LR}}(t, x_1, p_1)\Big|_{t \to 0^+} =$$
$$= \frac{a}{(2\pi)^2}1^{\mathrm{dr}}(x_1)\int \mathrm{d}p_2 (p_1 - p_2)C^{\mathrm{n}}(x_1, p_1, x_1 + (p_1 - p_2)0^+, p_2). \quad \text{(A.33)}$$

Using the monotonicity of the effective velocity this can be brought into the form (45).

# B Conditioning of initial measure

For the derivation in Appendix A we need to compute the expectation value of various observables conditioned on the already fixed position of $m$ particles $z_1 = (x_1, p_1), \ldots, z_n = (x_n, p_n)$. Here we are going to derive simplified expressions for these using the large deviation principle. The starting point is the following expansion of the expectation of product of observables $A_1, \ldots, A_n$ in terms of cumulants:

$$\langle A_1 \ldots A_n \rangle = \sum_\pi \prod_{B \in \pi} \kappa(\{A_i, i \in B\}). \tag{B.1}$$

Here $\pi$ runs over all partitions of $\{1, \ldots, n\}$, $B$ runs over each set in the partitioning and $\kappa(A_1, A_2, \ldots)$ is the joint cumulant of $A_1, A_2, \ldots$. In the usual LD scaling $\kappa(A_i) = \langle A_i \rangle = \mathcal{O}(1)$, $\kappa(A_i, A_j) = \text{Cov}[A_i, A_j] = \langle A_i A_j \rangle^c = \mathcal{O}(1/\ell)$, $\kappa(A_i, A_j, A_k) = \mathcal{O}(1/\ell^2)$ and so on. Since we are only interested in terms including $1/\ell$ we can write:

$$\langle A_1 \ldots A_n \rangle = \prod_i \langle A_i \rangle + \sum_{i<j} \langle A_i A_j \rangle^c \prod_{k \neq i,j} \langle A_k \rangle + \mathcal{O}(1/\ell^2). \tag{B.2}$$

Let us now apply this to $A_i = \rho_e(z_i)$, where $z_i = (x_i, p_i)$ (a more precise way would be to integrate $\rho_e(z)$ against a test function):

$$\langle \rho_e(z_1) \ldots \rho_e(z_n) \rangle = \prod_i \langle \rho_e(z_i) \rangle + \sum_{i<j} \langle \rho_e(z_i) \rho_e(z_j) \rangle^c \prod_{k \neq i,j} \langle \rho_e(z_k) \rangle + \mathcal{O}(1/\ell^2) =$$
$$= \prod_i \langle \rho_e(z_i) \rangle + \sum_{i<j} (f_2^c(z_i, z_j) + \tfrac{1}{\ell} \delta(z_i - z_j) \langle \rho_e(z_i) \rangle) \prod_{k \neq i,j} \langle \rho_e(z_k) \rangle + \mathcal{O}(1/\ell^2), \tag{B.3}$$

where we denoted

$$f_2^c(z_1; z_2) = f_2(z_1; z_2) - \langle \rho_e(z_1) \rangle \langle \rho_e(z_2) \rangle. \tag{B.4}$$

In general, we denote the $n$'th marginal distribution $f_n(z_1, \ldots, z_n)$ by

$$f_n(z_1; \ldots; z_n) = \left\langle \frac{1}{\ell^n} \sum_{1 \leq i_1, \ldots, i_n \leq N : i_k \neq i_l} \prod_{k=1}^n \delta(z_k - z_{i_k}) \right\rangle. \tag{B.5}$$

Note that we can write

$$\langle \rho_e(z_1) \ldots \rho_e(z_n) \rangle = \left\langle \frac{1}{\ell^n} \sum_{1 \leq i_1, \ldots, i_n \leq N} \prod_{k=1}^n \delta(z_k - z_{i_k}) \right\rangle =$$
$$= f_n(z_1; \ldots; z_n) + \frac{1}{\ell} \sum_{1 \leq i < j \leq n} \delta(z_i - z_j) f_{n-1}(z_1; \ldots; \hat{z}_i; \ldots; z_n) + \mathcal{O}(1/\ell^2) = \tag{B.6}$$
$$= f_n(z_1; \ldots; z_n) + \frac{1}{\ell} \sum_{1 \leq i < j \leq n} \delta(z_i - z_j) \prod_{k \neq i} \langle \rho_e(z_k) \rangle + \mathcal{O}(1/\ell^2).$$

Here $\hat{z}_i$ means that we omit $z_i$. In the last step we used that the dominant contribution to $f_n(z_1; \ldots; z_n)$ is simply given by the product of the individual densities.

Comparing this expression to (B.3) we conclude

$$f_n(z_1; \ldots; z_n) = \prod_i \langle \rho_e(z_i) \rangle + \sum_{i<j} f_2^c(z_i; z_j) \prod_{k \neq i,j} \langle \rho_e(z_k) \rangle + \mathcal{O}(1/\ell^2). \tag{B.7}$$

It also follows

$$f_{n+1}(z_1; \ldots; z_{n+1}) = f_n(z_1; \ldots; z_n) \langle \rho_e(z_{n+1}) \rangle +$$
$$+ \frac{1}{\ell} \sum_{i=1}^{n} f_2^c(z_i; z_{n+1}) \prod_{k \neq i} \langle \rho_e(z_k) \rangle + \mathcal{O}(1/\ell^2), \quad \text{(B.8)}$$

which finally gives

$$\frac{f_{n+1}(z_1; \ldots; z_{n+1})}{f_n(z_1; \ldots; z_n)} = \langle \rho_e(z_{n+1}) \rangle + \frac{1}{\ell} \sum_{i=1}^{n} \frac{f_2^c(z_i; z_{n+1})}{\langle \rho_e(z_i) \rangle} + \mathcal{O}(1/\ell^2). \tag{B.9}$$

This last result tells us what the distribution of a single particle $z_{n+1}$ is, given that $n$ other particles $z_1, \ldots, z_n$ are already known.

We can use this to write the expectation value of $n_{(x_1, x_2)}$ (the number of particles between $x_1, x_2$) conditioned on the presence of other particles as follows:

$$\langle n_{(x_1, x_2)} | z_1; \ldots; z_n \rangle = \int_{x_1}^{x_2} dx_{n+1} \, dp_{n+1} \frac{f_{n+1}(z_1; \ldots; z_{n+1})}{f_n(z_1; \ldots; z_n)} + \frac{1}{\ell} \sum_{i=3}^{n} \mathbf{1}_{(x_1, x_2)}(x_i). \tag{B.10}$$

Here $\mathbf{1}_{(x_1, x_2)}(x)$ is the (signed) indicator function on the interval $(x_1, x_2)$, with negative sign if $x_2 < x_1$. We can insert (B.9) into this, finding

$$\langle n_{(x_1, x_2)} | z_1; \ldots; z_n \rangle = \langle n_{(x_1, x_2)} \rangle +$$
$$+ \frac{1}{\ell} \sum_{i=1}^{n} \int_{x_1}^{x_2} dx_{n+1} \, dp_{n+1} \frac{f_2^c(z_i; z_{n+1})}{\langle \rho_e(z_i) \rangle} + \frac{1}{\ell} \sum_{i=3}^{n} \mathbf{1}_{(x_1, x_2)}(x_i) + \mathcal{O}(1/\ell^2). \tag{B.11}$$

Similarly we can compute:

$$\text{Var}[n_{(x_1, x_2)} | z_1, \ldots, z_n] = \int_{x_1}^{x_2} dx_{n+1} \, dx_{n+2} \, dp_{n+1} \, dp_{n+2} \frac{f_{n+2}(z_1; \ldots; z_{n+1}; z_{n+2})}{f_n(z_1; \ldots; z_n)} +$$
$$+ \frac{1}{\ell} \text{sgn}(x_2 - x_1) \int_{x_1}^{x_2} dx_{n+1} \, dp_{n+1} \frac{f_{n+1}(z_1; \ldots; z_{n+1})}{f_n(z_1; \ldots; z_n)} - \langle n_{(x_1, x_2)} | z_1; \ldots; z_n \rangle^2$$
$$= \int_{x_1}^{x_2} dx_{n+1} \, dx_{n+2} \, dp_{n+1} \, dp_{n+2} \frac{f_{n+2}(z_1; \ldots; z_{n+1}; z_{n+2})}{f_{n+1}(z_1; \ldots; z_{n+1})} \frac{f_{n+1}(z_1; \ldots; z_{n+1})}{f_n(z_1; \ldots; z_n)} +$$
$$+ \frac{1}{\ell} \text{sgn}(x_2 - x_1) \int_{x_1}^{x_2} dx_{n+1} \, dp_{n+1} \frac{f_{n+1}(z_1; \ldots; z_{n+1})}{f_n(z_1; \ldots; z_n)} - \langle n_{(x_1, x_2)} | z_1; \ldots; z_n \rangle^2 \tag{B.12}$$
$$= \int_{x_1}^{x_2} dx_{n+1} \, dx_{n+2} \, dp_{n+1} \, dp_{n+2} \, f_2^c(z_{n+1}; z_{n+2}) +$$
$$+ \frac{1}{\ell} \text{sgn}(x_2 - x_1) \int_{x_1}^{x_2} dx_{n+1} \, dp_{n+1} \langle \rho_e(z_{n+1}) \rangle + \mathcal{O}(1/\ell^2)$$
$$= \text{Var}[n_{(x_1, x_2)}] + \mathcal{O}(1/\ell^2).$$

Note that this result does not depend on the conditioning. The same is true for the covariances:

$$\text{Cov}[n_{(x_1, x_2)}, n_{(x_1, x_3)} | z_1; \ldots; z_n] = \int_{x_1}^{x_2} dx_{n+1} \int_{x_1}^{x_3} dx_{n+2} \, dp_{n+1} \, dp_{n+2} \, f_2^c(z_{n+1}, z_{n+2}) +$$
$$+ \frac{1}{\ell} \theta((x_2 - x_1)(x_3 - x_1)) \int_{x_1 \wedge (x_2 \vee x_3)}^{x_1 \vee (x_2 \wedge x_3)} dx_{n+1} \, dp_{n+1} \langle \rho_e(z_{n+1}) \rangle + \mathcal{O}(1/\ell^2) \tag{B.13}$$
$$= \text{Cov}[n_{(x_1, x_2)}, n_{(x_1, x_3)}] + \mathcal{O}(1/\ell^2).$$

## C  Simplified formulas for constant particle density

In the case of constant particle density, which is the case considered in [14], the formulas can be simplified considerably. In this case the initial state is given by:

$$\rho(0, x, p) = \bar{\rho} h(x, p), \tag{C.1}$$

where $h(x, p)$ is the local momentum distribution, i.e. $\int \mathrm{d}p\, h(x, p) = 1$, and with correlations only given by the GGE part (20), i.e. $C_{\mathrm{LR}}(x, p, y, q) = 0$. Note that $1^{\mathrm{dr}}$ is independent of $x$ and $\Gamma(x) = \bar{\rho} 1^{\mathrm{dr}^2} x + c$. Furthermore we have

$$\hat{X}(x) = x - a\bar{\rho}x + c = 1^{\mathrm{dr}} x + c, \tag{C.2}$$

from which we find

$$X(\hat{X}(x) + (p_1 - p_2)t) = x + \frac{p_1 - p_2}{1^{\mathrm{dr}}} t. \tag{C.3}$$

Therefore we can simplify

$$X(t, x_1, p_1) = x_1 + p_1 t + a\bar{\rho} \int \mathrm{d}x_2 \, \mathrm{d}p_2 \, h(x_2, p_2)(\theta(x_1 - x_2 + \tfrac{p_1 - p_2}{1^{\mathrm{dr}}} t) - \theta(x_1 - x_2)), \tag{C.4}$$

and

$$
\begin{aligned}
\Delta X_{\mathrm{local}}(t, x_1, p_1) = {} & a^2 1^{\mathrm{dr}} \bar{\rho} \int \mathrm{d}p_2 \, h(x_1 + \tfrac{p_1 - p_2}{1^{\mathrm{dr}}} t, p_2) \operatorname{sgn}(p_1 - p_2) + \\
& + \tfrac{a^3}{2} \bar{\rho}^2 \int \mathrm{d}p_2 \, \partial_x h(x_1 + \tfrac{p_1 - p_2}{1^{\mathrm{dr}}} t, p_2) \tfrac{|p_1 - p_2|}{1^{\mathrm{dr}}} t + \\
& + \tfrac{a^3}{2} \bar{\rho}^2 \int \mathrm{d}p_2 \, h(x_1 + \tfrac{p_1 - p_2}{1^{\mathrm{dr}}} t, p_2) \operatorname{sgn}(p_1 - p_2) + \\
& + \tfrac{a}{2} \bar{\rho} \int \mathrm{d}p_2 \, h(x_1, p_2)(-2a + a^2 \bar{\rho}) \operatorname{sgn}(p_1 - p_2)
\end{aligned}
\tag{C.5}
$$

and

$$
\begin{aligned}
V_{\mathrm{local}}(t, x_1, p_1) = {} & 2a^3 \bar{\rho}^3 1^{\mathrm{dr}} \int \mathrm{d}x_2 \, \mathrm{d}p_2 \, \mathrm{d}p_3 \, h(x_2, p_2) h(x_3, p_3) \times \\
& \times (\theta(x_1 - x_2 + \tfrac{p_1 - p_2}{1^{\mathrm{dr}}} t) - \theta(x_1 - x_2)) \mathbf{1}_{(x_1, x_3)}(x_2) \Big|_{x_3 = x_1 + \frac{p_1 - p_3}{1^{\mathrm{dr}}} t} + \\
& + a^4 \bar{\rho}^3 \int \mathrm{d}p_2 \, \mathrm{d}p_3 \, h(x_1, p_2) h(x_1, p_3) \theta((p_1 - p_2)(p_1 - p_3)) \tfrac{|p_1 - p_2| \wedge |p_1 - p_3|}{1^{\mathrm{dr}}} t + \\
& + a^2(-2a + a^2 \bar{\rho}) \bar{\rho}^2 \int \mathrm{d}x_2 \left[ \int \mathrm{d}p_2 \, h(x_2, p_2)(\theta(x_1 - x_2 + \tfrac{p_1 - p_2}{1^{\mathrm{dr}}} t) - \theta(x_1 - x_2)) \right]^2 + \\
& + a^2 \bar{\rho} \int \mathrm{d}x_2 \, \mathrm{d}p_2 \, h(x_2, p_2) \left[ \theta(x_1 - x_2 + \tfrac{p_1 - p_2}{1^{\mathrm{dr}}} t) - \theta(x_1 - x_2) \right]^2.
\end{aligned}
\tag{C.6}
$$

## D  Evolution of the correlation function

The general evolution formula for Euler scale correlation functions in a general integrable model has been derived recently [25–27]. For completeness we will give a quick derivation

here in the hard rods case. First we define the height field:

$$\Phi(t,x,p) = \int_{-\infty}^{x} \mathrm{d}y\, \rho(t,y,q). \tag{D.1}$$

Now we can define the time-dependent contracted space-coordinate

$$\hat{X}(t,x) = x - a \int \mathrm{d}p\, \Phi(t,x,p). \tag{D.2}$$

Note that from the GHD equation one obtains:

$$\frac{\mathrm{d}}{\mathrm{d}t}\hat{X}(t,x) = a \int \mathrm{d}p\, v^{\mathrm{eff}}(t,x,p)\rho(t,x,p) = a \int \mathrm{d}p\, p\rho(t,x,p). \tag{D.3}$$

We can also define the height field in contracted coordinates

$$\hat{\Phi}(t,\hat{X}(t,x),p) = \Phi(t,x,p), \tag{D.4}$$

and $\hat{\rho}(t,\hat{x},p) = \partial_{\hat{x}}\hat{\Phi}(t,\hat{x},p)$ and by taking the time-derivative we find

$$\partial_t \hat{\Phi}(t,\hat{X}(t,x,p),p) + a\hat{\rho}(t,\hat{X}(t,x,p),p) \int \mathrm{d}p\, p\rho(t,x,p) = -v^{\mathrm{eff}}(t,x,p)\rho(t,x,p) \tag{D.5}$$

Simplifying this one obtains

$$\partial_t \hat{\Phi}(t,\hat{x},p) + p\partial_x \hat{\Phi}(t,\hat{x},p) = 0, \tag{D.6}$$

and thus $\hat{\Phi}(t,\hat{x},p) = \hat{\Phi}(0,\hat{x}-pt,p)$ as it should because the evolution in contracted coordinates is free.

    Therefore the algorithm to evolve the Euler scale correlation functions is as follows. Given an initial correlation function $\langle \delta\rho_{\mathrm{e}}(0,x,p)\delta\rho_{\mathrm{e}}(0,y,q)\rangle$ first compute the height field correlation function

$$\langle \delta\Phi(0,x,p)\delta\Phi(0,y,q)\rangle = \int_{-\infty}^{x} \mathrm{d}x' \int_{-\infty}^{y} \mathrm{d}y' \langle \delta\rho_{\mathrm{e}}(0,x',p)\delta\rho_{\mathrm{e}}(0,y',q)\rangle. \tag{D.7}$$

Next we will use

$$\delta\hat{\Phi}(0,\hat{X}(0,x),p) - a\hat{\rho}(0,\hat{X}(0,x),p) \int \mathrm{d}q\, \delta\Phi(0,x,q) = \delta\Phi(0,x,p), \tag{D.8}$$

which can be solved as

$$\delta\Phi(0,\hat{X}(0,x),p) = \int \mathrm{d}q\, (\delta(p-q) + \tfrac{a}{2\pi}n(0,x,p))\delta\Phi(0,x,q), \tag{D.9}$$

to obtain the contracted height field correlation functions

$$\langle \delta\hat{\Phi}(0,\hat{X}(0,x),p)\delta\hat{\Phi}(0,\hat{X}(0,y),q)\rangle = \int \mathrm{d}p'\,\mathrm{d}q'\, (\delta(p-p') + \tfrac{a}{2\pi}n(0,x,p))\times$$
$$\times (\delta(q-q') + \tfrac{a}{2\pi}n(0,y,q))\langle \delta\Phi(0,x,q)\delta\Phi(0,y,q')\rangle. \tag{D.10}$$

These correlations evolve trivially in time

$$\langle \delta\hat{\Phi}(t,\hat{X}(0,x)+pt,p)\delta\hat{\Phi}(t,\hat{X}(0,y)+qt,q)\rangle = \langle \delta\hat{\Phi}(0,\hat{X}(0,x),p)\delta\hat{\Phi}(0,\hat{X}(0,y),q)\rangle. \tag{D.11}$$

As a last step we need to change from contracted coordinates to physical coordinates

$$\delta\Phi(t,\hat{X}^{-1}(t,\hat{x}),p) + a\rho(t,\hat{X}^{-1}(t,\hat{x}),p)\int dq\,\delta\hat{\Phi}(t,\hat{x},q) = \delta\hat{\Phi}(t,\hat{x},p), \tag{D.12}$$

from which we find

$$\delta\Phi(t,x,p) = \int dq\,(\delta(p-q) - a\rho(t,x,p))\delta\hat{\Phi}(t,\hat{X}(t,x),q). \tag{D.13}$$

Thus we find

$$
\begin{aligned}
&\langle\delta\Phi(t,x,p)\delta\Phi(t,y,q)\rangle \\
&= \int dp'\,dq'\,(\delta(p-p') - a\rho(t,x,p))(\delta(q-q') - a\rho(t,y,q))\times \\
&\quad \times\langle\delta\hat{\Phi}(t,\hat{X}(t,x),p')\delta\hat{\Phi}(t,\hat{X}(t,y),q')\rangle \\
&= \int dp'\,dq'\,(\delta(p-p') - a\rho(t,x,p))(\delta(q-q') - a\rho(t,y,q))\times \\
&\quad \times\int dp''\,dq''\,(\delta(p'-p'') + \tfrac{a}{2\pi}n(0,x',p'))(\delta(q'-q'') + \tfrac{a}{2\pi}n(0,y',q'))\times \\
&\quad \times\langle\delta\Phi(0,x',p'')\delta\Phi(0,y',q'')\rangle\bigg|_{x'=\hat{X}^{-1}(0,\hat{X}(t,x)-p't),\,y'=\hat{X}^{-1}(0,\hat{X}(t,y)-q't)}.
\end{aligned}
\tag{D.14}
$$

Finally we can obtain $\langle\delta\rho_e(t,x,p)\delta\rho_e(t,y,q)\rangle$ via

$$\langle\delta\rho_e(t,x,p)\delta\rho_e(t,y,q)\rangle = \partial_x\partial_y\langle\delta\Phi(t,x,p)\delta\Phi(t,y,q)\rangle. \tag{D.15}$$

## D.1 Correlations starting from local GGE state

In a local GGE state the correlations are given by (20). Therefore at $t=0$:

$$\langle\delta\Phi(0,x,p)\delta\Phi(0,y,q)\rangle = \int_{-\infty}^{x\wedge y} dx'\,\rho(0,x',p)\delta(p-q) + \gamma(x',p,q). \tag{D.16}$$

Therefore when applying two derivatives on (D.14) we have multiple options. If both derivatives act on one of the smooth $\rho(t,x,p)$ or $n(0,x,p)$ we will obtain a continuous function. If both derivatives act on the initial correlations we will obtain a $\delta(x-y)$ function and if only one of them acts on the initial correlations we will obtain a step function. This gives:

$$
\begin{aligned}
&\ell\,\langle\delta\rho_e(t,x,p)\delta\rho_e(t,y,q)\rangle\bigg|_{y\approx x} = \\
&= \delta(x-y)\big[\rho(t,x,p)\delta(p-q) + \rho(t,x,p)\rho(t,x,q)(-2a+a^2\bar{\rho}(t,x))\big] + \\
&\quad + \tfrac{a}{2}1^{dr}(t,x)[\partial_x\rho(t,x,p)\rho(t,x,q) - \rho(t,x,p)\partial_x\rho(t,x,q)]\,\mathrm{sgn}(y-x) + \\
&\quad + (\text{continuous}).
\end{aligned}
\tag{D.17}
$$

We can also write this in terms of the occupation function:

$$
\begin{aligned}
&\langle\delta n(t,x,p)\delta n(t,y,q)\rangle\bigg|_{y\approx x} = \tfrac{(2\pi)^2}{1^{dr}(t,x)^2}\delta(x-y)\rho(t,x,p)\delta(p-q) + \\
&\quad + \tfrac{a(2\pi)^2}{21^{dr}(t,x)^2}[\partial_x\rho(t,x,p)\rho(t,x,q) - \rho(t,x,p)\partial_x\rho(t,x,q)]\,\mathrm{sgn}(y-x) + \\
&\quad + (\text{continuous}).
\end{aligned}
\tag{D.18}
$$

The continuous part has a simple expression for short times $t \to 0^+$. In fact we have for $y = x + vt$:

$$\langle \delta n(t,x,p)\delta n(t,x+vt,q)\rangle = \frac{(2\pi)^2}{t 1^{\mathrm{dr}}(0,x)^2}\delta(v)\rho(x,p)\delta(p-q)+$$
$$+ a(\theta(v - v^{\mathrm{eff}}(0,x,q) + v^{\mathrm{eff}}(0,x,p)) - \theta(v))n(0,x,p)\partial_x n(0,x,q)+ \quad \text{(D.19)}$$
$$+ a(\theta(-(v - v^{\mathrm{eff}}(0,x,q) + v^{\mathrm{eff}}(0,x,p))) - \theta(-v))n(0,x,q)\partial_x n(0,x,p) + \mathcal{O}(t).$$

We can read off

$$C^{\mathrm{n}}_{\mathrm{LR,sym}}(t = 0^+ x,p,q) =$$
$$= \frac{a}{2}\,\mathrm{sgn}(p-q)[n(0,x,p)\partial_x n(0,x,q) - n(0,x,q)\partial_x n(0,x,p)] = \quad \text{(D.20)}$$
$$= \frac{a(2\pi)^2}{2 1^{\mathrm{dr}}(t,x)^2}\,\mathrm{sgn}(p-q)[\rho(0,x,p)\partial_x \rho(0,x,q) - \rho(0,x,q)\partial_x \rho(0,x,p)].$$

# E   Relation to theory of diffusion in general integrable systems

In our companion paper [1] we propose a general theory for diffusion in models with so called linearly degenerate hydrodynamics (which includes integrable models). The main equation there is [1, Eq. (9)], which for hard rods reads:

$$\partial_t \rho(x,p) + \partial_x(v^{\mathrm{eff}}(x,p)\rho(x,p))$$
$$+ \frac{1}{2L}\partial_x\left(\int dq_1\, dq_2\, \frac{\delta j(t,x,p)}{\delta\rho(q_1)\delta\rho(q_2)}C_{\mathrm{LR,sym}}(x,q_1,q_2)\right) = 0. \quad \text{(E.1)}$$

Here $j(p) = v^{\mathrm{eff}}(p)\rho(p)$. One can explicitly compute (suppressing the $x$ argument for convenience)

$$\frac{\delta j(p)}{\delta\rho(q_1)\delta\rho(q_2)} = -\frac{a}{1^{\mathrm{dr}}}(q_2\delta(p-q_1) + q_1\delta(p-q_2)) + a\frac{v^{\mathrm{eff}}(p)}{1^{\mathrm{dr}}}(\delta(p-q_1) + \delta(p-q_2))$$
$$- \frac{a^2}{1^{\mathrm{dr}2}}(q_1 + q_2)\rho(p) + \frac{2a^2 v^{\mathrm{eff}}(p)}{1^{\mathrm{dr}2}}\rho(p). \quad \text{(E.2)}$$

Integrating this against the correlation function and using the fact that this is symmetric under $p \leftrightarrow q$, we have:

$$\int dq_1\, dq_2\, \frac{\delta j(p)}{\delta\rho(q_1)\delta\rho(q_2)}C_{\mathrm{LR,sym}}(q_1,q_2) = 2\int dq_1\, dq_2\left(-\frac{a}{1^{\mathrm{dr}}}q_2\delta(p-q_1)+\right.$$
$$+ a\frac{v^{\mathrm{eff}}(p)}{1^{\mathrm{dr}}}\delta(p-q_1) - \frac{a^2}{1^{\mathrm{dr}2}}q_2\rho(p) + a^2\frac{v^{\mathrm{eff}}(p)}{1^{\mathrm{dr}2}}\rho(p)\Big)C_{\mathrm{LR,sym}}(q_1,q_2)$$
$$= \frac{2a}{1^{\mathrm{dr}}}\int dq_1\, dq_2\left(\delta(p-q_1) + a\frac{\rho(p)}{1^{\mathrm{dr}}}\right)(-q_2 + av^{\mathrm{eff}}(p))C_{\mathrm{LR,sym}}(q_1,q_2) \quad \text{(E.3)}$$
$$= \frac{2a}{1^{\mathrm{dr}}}\int dq\,(p-q)\int dq_1\, dq_2\left(\delta(p-q_1) + a\frac{\rho(p)}{1^{\mathrm{dr}}}\right)\left(\delta(q_2-q) + a\frac{\rho(q)}{1^{\mathrm{dr}}}\right)C_{\mathrm{LR,sym}}(q_1,q_2).$$

This is equivalent to (44) and by following the steps towards (51) one can easily see that (E.1) is equivalent to (51). Hence we have shown that the general equation in [1] reduces to our equation in the hard rods case.

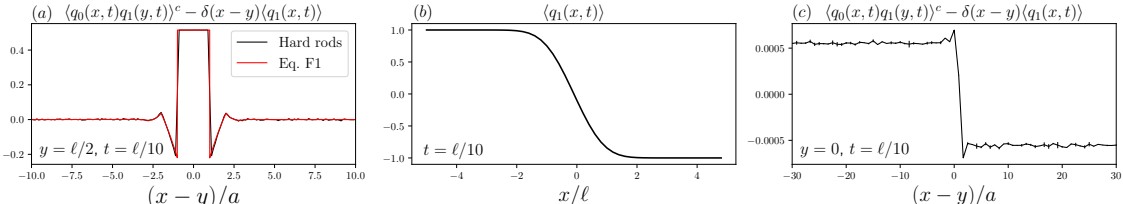

Figure 6: (a) Plot for the microscopic correlations in a Hard rods gas, with rods length $a = 0.3$ and scale $\ell = 200$. Respectively, from left to right, we plot $\langle q_0(x,t)q_1(y,t)\rangle^c_{\mathrm{reg}} - \delta(x-y)\langle q_1(x,t)\rangle$ at the points $y = \ell/2$, $t = \ell/10$ and $\ell = 200$. The Hard rods data (black line) is compared with prediction from Eq. (F.1) (red line). The Hard rod data are averaged over $10^7$ initial states. Eq. (F.1) has been evaluated truncating the summation at $k = 50$, such that the truncation error is $< 10^{-16}$. The agreement between the prediction and the numerical data is excellent, up to the Hard rods monte carlo noise. We stress that discrepancies induced by long-range correlations are expected to be order $\mathcal{O}(\ell^{-1})$. (b) Plot of $\langle q_1(x,t)\rangle$ at a time $t = \ell/10$ from Hard rods numerics. The figure (c) shows the microscopic correlations $\langle q_0(x,t)q_1(y,t)\rangle^c_{\mathrm{reg}} - \delta(x-y)\langle q_1(x,t)\rangle$ at $y = 0$ and $t = 10/\ell$. At this point, the leading contribution in $\ell$ is expected to be vanishing, having $\langle q_0(x = 0, t = \ell/10)\rangle \simeq 0$ (as shown in box (b)), and since $\langle q_0(x,t)q_1(y,t)\rangle^c_{\mathrm{reg}} \propto \langle q_1(x,t)\rangle + \mathcal{O}(\ell^{-1})$. Hence, this plots shows the microscopic structure of the discontinuity in the two-point correlation. We can conclude that the 'jump' is developed at a microscopic length scale $\sim a$. For numerical simulation we used the same data as [1].

# F   Microscopic correlations in the hard rod gas

In this section, we show numerical results for the microscopic correlation functions in a hard rods gas. Hence, in this section we refer to $x$ as to the hard rods microscopic position. Let us consider a homogeneous hard rods gas, with rods' length $a$ and particle density $\rho(p) = \bar{\rho}h(p)$, with $\int dp\, h(p) = 1$ and $p$ the particles' momentum. The microscopic correlation function for such a system is given by [28]

$$C^{\mathrm{micro}}(x,p,x',p') = \delta(x-x')\delta(p-p')\bar{\rho}h(p) + (\mathfrak{n}^{(2)}(x-x') - \bar{\rho}^2)h(p)h(p') \qquad \text{(F.1)}$$

$$\mathfrak{n}^{(2)}(x) = \frac{\bar{\rho}^2}{1-\bar{\rho}a}\sum_{k=1}^{\infty}\frac{1}{(k-1)!}\left(\frac{|x|-ak}{\bar{\rho}^{-1}-a}\right)^{k-1}\exp\left[-\left(\frac{|x|-ak}{\bar{\rho}^{-1}-a}\right)\right]\theta(|x|-ka), \qquad \text{(F.2)}$$

where $\theta$ is the Heaviside step function. Let us now consider a non-uniform gas, being $\ell$ the typical scale of spatial variations. Its correlations will be given by $\langle \rho(x,p)\rho(x,p')\rangle^c = C^{\mathrm{micro}}(x,p,x',p') + \mathcal{O}(\ell^{-1})$, where $C^{\mathrm{micro}}(x,p,x',p')$ is defined by taking $\bar{\rho} \to \bar{\rho}(x,t)$ with $\bar{\rho}(x,t) \equiv \int dp\, \rho(x,p,t)$ and $h(p) \to h_x(p,t) \equiv \rho(x,p,t)/\bar{\rho}(x,t)$. We also stress that here $(x,x')$ are microscopic spatial coordinates.

We compare the numerical results for the microscopic correlations functions of a time evolving hard rod gas with the prediction of Eq. (F.1). More precisely, we consider a gas of hard rods with only two modes, $p_+ = +1$ and $pa_- = -1$, evolving from the initial state $\rho_{\pm}(x,t=0) = 1 \mp \mathrm{Erf}(x/\ell)/2$. We also define the quatities $q_0(x) \equiv \rho_+(x) + \rho_-(x)$ and $q_1(x) \equiv \rho_+(x) - \rho_-(x)$.

Hence, in the initial state $q_0(x,t=0) = \rho_+(x,t=0) + \rho_-(x,t=0) = 1$ and $q_1(x,t=0) = \rho_+(x,t=0) - \rho_-(x,t=0) = -\mathrm{Erf}(x/\ell)$. In Fig. 6(a) we show the hard-rod numerical results for $\langle q_0(x,t)q_1(y,t)\rangle^c - \delta(x-y)\langle q_1(x,t)\rangle$ at the points $y = \ell/2$ and $t = \ell/10$ for $\ell = 200$. Comparing it with the theoretical predictions from Eq. (F.1), we observe excellent

agreement. We also stress that all the discrepancies induced by the long-range correlations are at order $\mathcal{O}(\ell^{-1})$, hence they are not visible in Fig. 6 (a). More precisely, we used the relation

$$
\langle q_0(x)q_1(y)\rangle^c = \sum_{i,j=1}^2 C^{\text{micro}}(x,p_i,y,p_j)p_i p_j + \mathcal{O}(\ell^{-1}),
$$

$$
\text{with:} \quad \sum_{i,j=1}^2 C^{\text{micro}}(x,p_i,y,p_j)p_i p_j \propto \langle q_1(y)\rangle,
$$

(F.3)

where we used $p_i = [-1,+1]$ and where $C^{\text{micro}}$ is defined in Eq. (F.1). But, as it is possible to observe from Fig. 6 (b), $\langle q_1(y,t)\rangle$ is vanishing at $t = \ell/10$ and $y = 0$. Thus, we expect the leading contribution to be vanishing at this point. This permits to isolate the long-range contribution to the correlations at microscopic scale and to observe the microscopic structure of the discontinuity introduced in the main text. In particular, in Fig. 6 (c), we show the two point function $\langle q_0(x,t)q_1(y,t)\rangle^c - \delta(x-y)\langle q_1(x,t)\rangle$ at $t = \ell/10$ and $y = 0$. From this picture, we can observe that the discontinuity predicted at hydrodynamic scale, is developed in the gas at scales $\sim a$.

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
