# Peer review of "Diffusive hydrodynamics of hard rods from microscopics"

_SciPost Physics_

## Round 2 · Referee Report · Anonymous (Referee 1) · 2025-7-1

Strengths

The manuscript provides an explicit microscopic derivation of hydrodynamics (HD) equations for hard rods that describe large-scale motion. These HD equations include diffusive effects, which appear as first-order corrections in the hydrodynamic gradient expansion. Starting from a local equilibrium GGE state exhibiting large deviation scaling with respect to a variation length scale \ell, the authors compute the evolution of the average of the empirical phase space density ρ(x, p) of the quasiparticles.

Inspecting the evolution of a quasiparticle starting from (x1, p1), they find that the average posi- tion ⟨Xe(t)|x1, p1)⟩ at order O(\ell^0) is given by the Euler characteristics. This is accompanied by two important corrections at O(1/\ell), originating from initial randomness and long-range correlations among the densities that develop over the Euler space-time scale. Incorporating these corrections into the phase space density ρ(x, p) of the quasiparticles, they find ‘diffusive’ corrections at O(1/\ell) to the Euler part of the current associated with the phase space density. Their explicit expression shows that this diffusive correction originates from the long-range correlations that evolve at the Euler space-time scale.

This result is strikingly different from previously derived Navier-Stokes (NS) hydrodynamics for hard rods. First, the NS equation is time-reversal asymmetric, while the new equation, coupled with the Euler equation of the two-point correlations, is time-reversal symmetric. Second, the NS equation applies to a local equilibrium state with no long-range correlation, whereas the equation derived in the current paper applies to general inhomogeneous states that can admit long-range correlations. I find the main result to be very interesting, the derivation novel, and it demonstrates an alternative mechanism of diffusion other than the Green-Kubo mechanism. It further makes important and significant improvement over the existing hydrodynamics of hard-rod gas beyond the Euler scale.

In summary:

  1. The paper obtains a diffusive scale hydrodynamics of hard rod gas that is strikingly different from previously derived Navier-Stokes equation. The paper clearly mentions at what condition the current result reduces to the NS equation.

  2. The paper provides a detailed derivation of diffusive scale hydrodynamics in a model starting from microscopic description which is generally a very hard problem.

  3. The paper demonstrates an interesting mechanism of diffusion originating from long range correlation at Euler scale.

Weaknesses

  1. The clarity of the paper at some places can be increased.

  2. The readability of the paper can be improved.

Report

The paper largely meets the journal's acceptance criterion. I would like to recommend publication of the paper. However, I feel the writing of the paper can be improved if a few things are clarified further.

Requested changes

  1. The new correction to the Euler GHD of hard rods is being called diffusive, although from equation (9) it is not obvious. However, its form in local equilibrium (eq. (3)) makes this clear. It would be great to clarify this point.
  2. The set of equations (9) and (55) being time-reversal symmetric, they do not show an `arrow of time’. Has this also been tested numerically with the definition of entropy in eq. (38) which does not include two-point correlation in its definition?
  3. Thecorrectiontotheaverageposition⟨Xe(t)|x1,p1)⟩ofthequasiparticletaggedbytheinitial position and velocity (x1, p1) receives two corrections at O(1/l): one from initial fluctuations and the other from long-range correlation. The former contribution is easy to understand. Is it possible to understand the second mechanism physically? Also, does the first contribution give rise to the Kubo diffusion (NS term) in Eq. (7)? Is the exact cancellation of the NS term and the Cn term universal? If possible, LR,asym it would be useful to make some intuitive comment on why such an exact cancellation occurs.
  4. Page (9) after Eq. (13): In order to obtain a macroscopic forward derivative in Eq. (9), one has to evaluate the RHS of eq. (9) at t = 0+ in macroscopic scale. This calculation has also been shown in eq. (D.20). Is it feasible to get the same solution by solving the coupled Eqs. (7) and (55) starting from a local GGE state, instead of the alternative computation presented in Appendix D?
  5. Would one find such a ’diffusion’ term originating from Cn LR well that support ballistic transport? in non-integrable systems as
  6. Typos: a. Eq. (11): It seems a t is missing from the argument ρe(..., y − veff ε, q), b. Eq. (55): ++ c. Caption of Fig. (3): It seems the reference to the theoretical prediction is not correct.
  7. I feel moving the discussion on X(t,x1,p1) along with Eqs. (34) and (35) immediately after Eq. (30) would be better for the readers.
  8. p12, after Eq. (38): I feel it would be useful for the reader to elaborate the following statement a little more: “However, because the structure of the state remains invariant under time and the hard rods dynamics do not have any memory, this immediately implies that (38) holds at all times.”

Attachment

Recommendation

Ask for minor revision

---

## Round 2 · Referee Report · Anonymous (Referee 2) · 2025-7-8

Strengths

See the attached PDF of my report

Weaknesses

See the attached PDF of my report

Report

See the attached PDF of my report

Requested changes

See the attached PDF of my report

Attachment

Recommendation

Ask for minor revision

---

## Round 2 · Referee Report · Anonymous (Referee 3) · 2025-8-23

Report on the paper *Diffusive hydrodynbamics of hard rods from microscopics* By F. Hubner, L Biagetti, J. De Nardis, B. Doyon.

The authors study the dynamics of a one-dimensional gas of hard rods, focusing in particular on the diffusive corrections to the ballistic macroscopic behaviour. They claim that the correct description of these corrections involves two coupled sets of equations: one governing the evolution of one-point functions (densities) and another for connected two-point correlations. Their main argument relies on the fact that the dynamics generates long-range correlations in the presence of density inhomogeneities. Previous derivations of diffusive corrections (so-called Navier–Stokes corrections) did not take such correlations into account.

The authors state (see Section 5, "Conclusions") that their derivation is not a rigorous mathematical proof, and should therefore be considered heuristic. However, I find too many mathematical imprecisions to be able to follow the argument in detail, even though the main motivation — that long-range correlations affect Navier–Stokes corrections — is sound. Below I list some of these imprecisions. I hope that another referee may be able to follow these heuristic calculations more closely. On the other hand, the agreement with numerical simulations suggests that the equations obtained may indeed be correct.

**Comments:**

- Page 4, Section 1.1: In stating the main results, the definition of the dynamics and the notation should be given beforehand. For example, in (5) what is $\rho_e$? How does this correlation depend on $\ell$?

- Page 7, Eq. (14): This is not a probability measure on the configuration space and cannot be normalized except under overly strong conditions on $\beta$. You probably mean a Poisson field with intensity $e^{-\beta}$, which should be stated precisely. (In Section 4 you mention this point correctly.)

- Page 7, 3rd line after (17): Who is $\ell^0$? Do you mean terms of order $\mathcal{O}(1)$?

- Page 7, Eq. (18): I do not see why you call this "large deviation scaling"; it appears instead to be a condition on the correlations of the fluctuation fields.

- Page 8, Assumption 2: If you include in the statement "In particular, this is satisfied for all time $t$...", it seems to be part of the assumption, whereas in Appendix D you show that it follows from the dynamics.

- Page 8, Assumption 3: I do not understand this assumption. Correlations are always symmetric by definition. What is meant by "the microscopic shape"?

- Page 12: It is not clear why (36) and (32) imply (37), since derivatives in $x$ are involved in (32). As the work is not a rigorous proof, this could perhaps be introduced as a reasonable assumption.

- Page 16, Eq. (60): With this density, the positions are not distributed according to a homogeneous Poisson process, as stated. The distances between them are therefore not identically distributed, but depend on their locations (they are rather localized under the density (60)).

---

## Editorial Decision

resubmitted